# Mesoscale Eddies in the Algerian Basin: do they differ as a function of their formation site?

Federica Pessini[1], Antonio Olita[1], Yuri Cotroneo[2], and Angelo Perilli[1]

[1]IAMC CNR di Oristano, Oristano, Italy
[2]Università degli Studi di Napoli "Parthenope", Napoli, Italy

*Correspondence to:* Federica Pessini (federica.pessini@iamc.cnr.it)

**Abstract.** The circulation of the Western Mediterranean Sea (WMED) is dominated by highly variable and heterogeneous mesoscale circulation, strongly driven by the formation and propagation of eddies (cyclonic and anticyclonic) mainly acting in the Algerian Basin.

In order to investigate the spatial and temporal distribution of eddy generation and their respective paths in the Algerian Basin, the most energetic WMED portion, we use an automated detection and tracking hybrid method applied to 24 years of Sea Level Anomaly (SLA) data. The algorithm is based on the computation of the Okubo-Weiss parameter in SLA closed loops and has been modified in order to fill the gaps in single eddy tracks. In this work we analyzed both cyclonic and anticyclonic structures, but the conclusions will be focused mainly on anticyclones with lifespan longer than three months, as they are characterized by higher kinetic energy, so potentially contributing to a large extent to the mesoscale characterization of the basin.

In particular, we find that anticyclonic short-life eddies mostly occur in the northern portion of the domain, north of $39°$ N, along the North Balearic Front (NBF). Such short-life eddies, labelled Frontal Eddies (FEs), are characterized by low translational velocity and a highly variable direction of propagation. We found a weak seasonality in their formation, with maxima in fall and winter. By contrast, anticyclonic longer-life eddies tend to arise in the southern part of the basin, along the Algerian Current, with a clear maximum in spring. All the structures (both cyclonic and anticyclonic) originating along the Algerian Current are known as Algerian Eddies (AEs). According to previous studies, we observe that these anticyclonic eddies mainly form east of $6.5°$ E and move eastward along the African coast to the Sardinia Channel, where they detach from the coast, continuing offshore and following the cyclonic intermediate circulation. We detect a region between $4.5°$ E and $6.5°$ E where such eddies tend to converge and terminate their life.

Finally, the analysis suggests that eddies formed in the northern and in the southern part of the Algerian Basin present some physical differences such as lifetime, kinetic energy and vorticity. Furthermore, the connection between the two parts in terms of eddy tracks is limited to a very small number of southbound (FEs) or northbound (AEs) structures crossing the $39°$ N.

## 1 Introduction

The Algerian Basin, located in the southern part of the Western Mediterranean Sea (WMED) between the African coast, the Balearic Islands and Sardinia is quite a small basin, characterized by both basin scale and mesoscale dynamics as also shown

in Figure 1, where the Mean Dynamic Topography (MDT) from Rio et al. (2014) is reported. The MDT is the permanent stationary component of the ocean dynamic topography and efficiently represents the main circulation of the basin. The Atlantic Water (AW) flows eastward from the Gibraltar Strait along the Algerian slope and forms the Algerian Current (AC). Past the Greenwich Meridian, the current becomes shallower and wider near the Sardinia Channel (Millot, 1985; Fusco et al., 2008, 2003). Mainly because of the difference in density between AW and the Mediterranean surrounding water (MW), this along-slope current frequently becomes unstable and meanders, generating mesoscale eddies as a result of baroclinic instabilities (Obaton et al., 2000). These structures can be both cyclonic and anticyclonic. In this work we will perform the analysis on both cyclones and anticyclones, with a focus on the latter. We will call Algerian Eddies (AEs) all the features formed along the Algerian Current, independently from their polarity (cyclonic or anticyclonic structures). They move along the African coast at a few Km per day and are blocked in proximity of the Sardinia Channel (Millot, 1985; Vignudelli, 1997; Millot, 1999; Font et al., 1998, 2004; Olita et al., 2011), where they are usually forced by the local topography to detach from the coast and follow a cyclonic pathway (Escudier et al., 2016a).

Mesoscale eddies in the western Mediterranean sea have been widely investigated in the past (Burkov et al., 1979) with in situ data (Benzhora and Millot, 1995b; Millot, 1999), moorings, surface drifters (Font et al., 2004), gliders (Amores et al., 2013; Cotroneo et al., 2016; Aulicino et al., 2016), altimetry observations (Isern-Fontanet et al., 2006; Escudier et al., 2016a; Mason and Pascual, 2013; Mason et al., 2014) and numerical models (Levy et al., 1999, 2000; Escudier et al., 2016b). Most qualitative information about the motion of the eddies are provided by infrared and colour satellite imagery (Millot, 1985), as most of the year, temperature gradients between the cores and the surrounding water are present. The biological response associated with eddies is remarkable and is usually highlighted by ocean colour signatures that could be used as a tracer (Moràn et al., 2001; Taupier-Letage et al., 2003; Olita et al., 2014). Infrared and ocean colour survey techniques are limited by the cloud cover, which does not allow continuous data recording, especially during winter. To avoid this limitation, mesoscale phenomena are investigated through altimetry observations, such as Absolute Dynamic Topography (ADT) and Sea Level Anomaly (SLA). They are not affected by cloud cover and provide the geostrophic velocity field, even if they are characterized by coarser spatial resolution than infrared and optical passive sensors.

The complexity of the dynamics in the Algerian Basin and its influence on the circulation of the entire WMED stimulated the study of the eddies and their motion. Puillat et al. (2002) analysed a long dataset of Sea Surface Temperature (SST) and observed AEs with lifespan up to 3 years. Henceforth, several automated detection and tracking algorithms applied to altimetry data have been developed. An analysis by Escudier et al. (2016a) based on SLA data in the Mediterranean reveals three preferred areas of formation for the AEs and also provides their mean pathways. In particular, structures formed in proximity to the Sardinia Channel remain close to the area of formation or detach from the coast, following the large-scale cyclonic Eastern Algerian Gyre (EAG). These features form most commonly during specific seasons and have significant inter-annual variability over the last 24 years.

In literature, other eddies of the Algerian Basin with different origins are described and labelled with several names, mainly as function of the formation area (more than on intrinsic eddy characteristics). Testor et al. (2005a) named Sardinian Eddies (SEs) the structures associated with the detachment of the EAG from the continental slope of Sardinia. Contrary to AEs, which

are baroclinic, these features are described by the Authors as strongly barotropic with a typical radius of about 30 Km. They also present a marked core of Levantine Intermediate Water (LIW) with high temperature and salinity at intermediate depths, with values similar to those found in the LIW vein along the Sardinian coast. SEs show a clear surface signature months after the detachment from the coast of Sardinia and their lifespans are shorter than one year.

The circulation of the Algerian Basin is separated from the cyclonic mean circulation of the northern Provençal Basin by the North Balearic Front (NBF), a thermal front characterized by a high seasonal variability (Ruiz et al., 2009; Olita et al., 2011). Typically, AW entering through the Strait of Gibraltar flows eastward at the Algerian Basin surface, while the water formed in the eastern and northern parts of the Mediterranean flows westward at the intermediate and deep layers (Millot, 1985; Aulicino et al., 2018). The surface layer presents large variability, while the deepest layers (below the LIW) are relatively uniform but

could experience short and long-term changes, as observed in Rixen et al. (2005); Fusco et al. (2008); Schroeder et al. (2016). An averaged west-east current, the Western Mid-Mediterranean Current (WMMC) (Pinardi et al., 2015), flows eastward from the Balearic Islands to the west coast of Sardinia in correspondence with the NBF. The front is affected by the seasonality of the north-westerly winds and of the mean circulation and reaches its southernmost position in winter. Along the current, baroclinic instabilities occur, probably contributing to the generation of meanders and eddies. From in situ data and satellite

data sets Fuda et al. (2000) observed several eddies propagating westward at a few Km/day along the NBF and named them Frontal Eddies (FEs). Most of them are short-lived and they usually present signature of Winter Intermediate Water (WIW) and LIW.

In spite of the current body of research, a comprehensive study of the tracking, energy and interaction of all kind of eddies forming and acting in the Algerian Basin is still lacking. In particular, the differences occurring among the eddies of different

origin are still not completely known, as the analysis of Escudier et al. (2016a) focused just on the AEs.

Our study aims to shed light on the different statistical characteristics (energy, path, formation areas, lifetime, etc.) of the two main types of eddies (AEs and FEs) in the Algerian Basin. This was done through a series of eulerian statistics (performed on a regular grid) computed on the output of the lagrangian tracking method adopted. The automated detection and tracking method used is a customized version of the hybrid method (combination of the physical and the geometrical approaches) performed by

Penven et al. (2005) and Halo et al. (2014), and it is detailed in the following section. Such method was applied on a long-term time series from 1993 to 2016 (24 years) of daily SLA fields, so providing a robust basis for statistical analysis.

Dataset, detection and tracking method are described in depth in Section 2, with some details in Appendix A. The results of our study are presented in Section 3. At first, we plotted the distribution of the eddies in the basin and their relative vorticity. Then we observed the distribution in time of the mean kinetic energy and its relation with the eddy lifetimes. We further computed

separate eulerian statistics of the formations and terminations of the structures for both short- and long-life eddies. Lastly, we observed the trajectories of the eddies and plotted the tracks and their translational velocity. In section 4 we summarize and discuss the results, and take our conclusions.

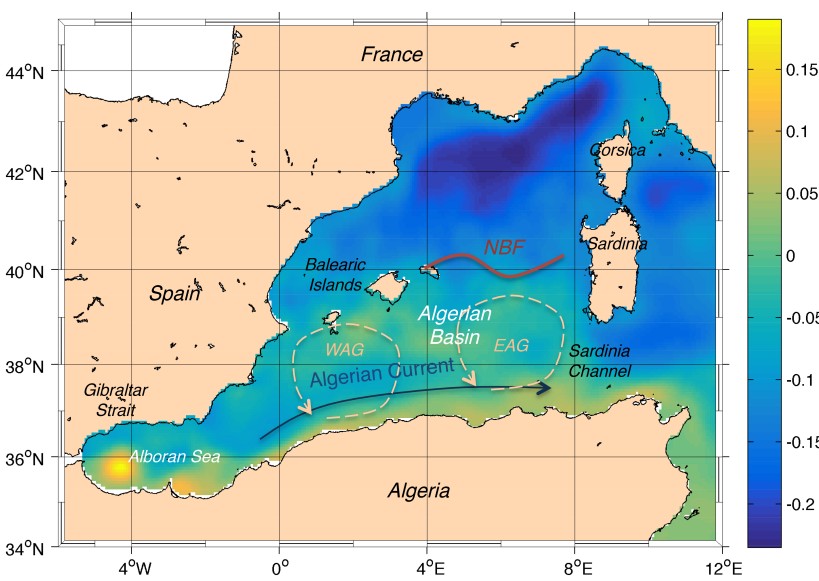

**Figure 1.** Mean Dynamic Topography (m) in the WMED. (Rio et al., 2014).

## 2 Materials and methods

Eddy detection and tracking algorithms based on altimetry data have been widely used to investigate mesoscale structures in both space and time (Isern-Fontanet et al., 2003; Chelton et al., 2007, 2011; Escudier et al., 2016a; Yi et al., 2014; Schütte et al., 2016; Mason and Pascual, 2013). These algorithms, applied on global and regional scales, have been successful in the characterization of several properties, such as polarity, pathway, lifetime, radius and amplitude. They can be applied to altimetry data sets or with realistic high-resolution numerical model outputs, which also offer the opportunity to study both surface and subsurface eddies, as well as their three-dimensional structures.

Many different methods have been developed for detecting eddies, but in general they fall into two main categories: physical and geometrical methods. The former are based on the calculation of some dynamic fields and define eddies using the closed contours of a threshold value. The physical fields computed are SLA, Sea Surface Height (SSH) (Chelton et al., 2011; Chaigneau et al., 2009), vorticity (McWilliams, 1999) and velocity gradient tensor or Okubo-Weiss parameter (W) (Penven et al., 2005; Chelton et al., 2007; Isern-Fontanet et al., 2003). On the other hand, geometrical methods are based on the curvature or shape of the instantaneous flow field (Sadarjoen and Post, 2000; Nencioli et al., 2010). Several studies make use of both physical and geometrical methods to better explain the structure of mesoscale eddies (Le Vu et al., 2017; Mkhinini et al., 2014; Halo et al., 2014).

In this study we used a hybrid method basing on the SLA closed contour criterion combined with the Okubo-Weiss parameter, which remains the most popular technique due to its simplicity and computational efficiency. After the detection, a tracking

algorithm identified the trajectories of the eddies and extrapolated several properties. In the next sections we describe the automated method of detection and tracking and the modifications we introduced to improve the results.

## 2.1 Sea Level Anomaly Data

The altimetry data used are the merged, delayed-time gridded maps of optimally interpolated SLA, previously available cour-
tesy of Archiving, Validation and Interpolation of Satellite Oceanographic (AVISO) and now distributed by Copernicus Marine Environment Monitoring Service (CMEMS). These data are produced by the SL-TAC multi-mission altimeter data processing system. The *all-sat* gridded SLA fields consider all available altimeters and therefore have higher quality levels, although not homogeneous in time due to the time-varying mission configuration. The horizontal resolution of the data is $1/8°$ (about 14 Km) and data are available from January 1993 to middle of 2017. This analysis covers the range $1993 - 2016$.
Details on the data, products and processing procedures are available on the SSALTO/DUACS User Handbook: www.aviso. altimetry.fr/fileadmin/documents/data/duacs/Duacs2014.pdf.

## 2.2 The detection and tracking algorithm

The Okubo-Weiss-based detection methods permit the separation of the cores of the eddies from the surrounding water on the basis of their physical properties. This method has proved its efficiency in separating the eddy core from the circulation
cell in terms of the sign of the Okubo-Weiss parameter (W) (Jeong and Hussain, 1995; Pasquero et al., 2001; Isern-Fontanet et al., 2003; Viikmae and Torsvik, 2013; Yi et al., 2014). Negative values of W correspond to the inner part of the eddy, the vorticity-dominated region, while positive values indicate the external part, dominated by strain. The borders of the structures are characterised by $W = 0$ (Chelton et al., 2007), while the areas of the structures found by this method correspond to the closed loops of the parameter W, which present negative values.
In this work we used a hybrid detection method (Halo et al., 2014): a combination of detecting the largest contours in SLA data through geometric criteria (Chelton et al., 2011) and computing the W parameter through a traditional physical method (Chelton et al., 2007). Three parameters can be tuned according to the area of study: the interval between the contours of SLA, the maximum radius of a closed contour of SLA detected and the threshold of the Okubo-Weiss parameter ($W_0$).
The eddy detection and tracking routines are coded in MatLab and are freely available for download thanks to Penven (2011).
We adapted the algorithm to our area of study, introducing some modifications and tuning the parameters. The algorithm has thus evolved thanks to many tests and was calibrated by comparison with independent satellite imagery (ocean colour and infrared, not shown).
The code can be divided in two main parts: the eulerian one, dedicated to the detection of the eddies (eddy detection), and the lagrangian one, which investigates their trajectories (eddy tracking) and has been modified to fill the gaps in single eddy track.
The hybrid method has been applied over the period from January 1993 to December 2016 to the area between $2°$ W and $10°$ E and from $36°$ N to $42°$ N.
The eulerian part concerns eddy detection starting from SLA data. After selecting a domain and a period of time the algorithm looks for the nearest dates of data available, then identifies the local maxima and minima that could eventually correspond to

anticyclonic and cyclonic eddies respectively. Then W is computed starting from geostrophic velocities. In order to reduce the grid scale noise, 2 passes of a Hanning filter are applied. The largest closed contours of negative W inside the SLA closed loops are considered to be an eddy. The closed loops of W are the border of the structures and correspond to the separation between the vorticity-dominated and the deformation-dominated region. The main steps of the detection algorithm and the discussion about the choice of the threshold value of the Okubo Weiss parameter are presented in Appendix A.

After the application of the detection routine, all the properties of the eddy such as center position, time, area, surface kinetic energy, vorticity, equivalent radius, maximum, minimum and mean sea surface height (SLA), amplitude, rotational speed and zonal and meridional propagation velocities are available.

The second part of the code is the lagrangian one, which provides information about the lifetime of the eddies and of their trajectories. Penven et al. (2005) proposed a method to track the eddies based on a generalized distance. After having selected the structures with correct properties, the algorithm detects an eddy in one altimetric snapshot and checks its presence in the subsequent frame, comparing each eddy with the structures of the following day. It takes into account the difference in coordinates, radius, vorticity, mean height and amplitude between two successive days. For each eddy $e_1$ of the initial day and for each eddy $e_2$ of the following day, $X_{e_1,e_2}$ is defined as a general distance in a non-dimensional property space:

$$ X_{e_1,e_2} = \left[ \left( \frac{L}{L_0} \right)^2 + \left( \frac{R_2 - R_1}{R_0} \right)^2 + \left( \frac{\xi_2 - \xi_1}{\xi_0} \right)^2 + \left( \frac{SLA_{mean_2} - SLA_{mean_1}}{Z_0} \right)^2 + \left( \frac{Amp_2 - Amp_1}{A_0} \right)^2 \right]^{\frac{1}{2}} \quad (1) $$

where $L$ is the spherical distance between the eddies and $R_2$, $R_1$, $\xi_2$, $\xi_1$, $SLA_{mean_2}$, $SLA_{mean_1}$, $Amp_2$, $Amp_1$ are respectively the radius, the vorticity, mean SLA variations and amplitude of $e_2$ and of $e_1$.

The eddy pair which minimizes the general distance is considered to be the same eddy evolving in time in all cases, except when the two following conditions are satisfied simultaneously: the eddy speed is greater than $U_{max} = 0.3$ m/s and the distance between the two structures ($L_0$) is greater than twice the radius of the first eddy ($> 2R_1$).

The algorithm counts an eddy when it is detected for the first time on a map of SLA and assigns an ID to facilitate the extrapolation of their properties. Eddies with the same ID are the same eddy evolving over time (eddy tracking). The method ensures that an eddy preserves its polarity, that is, a cyclonic or anticyclonic structure. No cyclones become anticyclones (and vice versa), and the translational speed remains in a realistic range of a few kilometers per day. If an eddy travels at an unrealistic translational speed, it is considered as a different eddy with a new ID.

We calculated for each eddy the date of formation, the date of termination, the lifetime, the translational velocity and the mean kinetic energy.

The main parameters calculated by the algorithm and the typical values, according to the dimension and to the properties of the Algerian Basin, are listed in Appendix A.

## 2.3 Detection algorithm changes

The altimetry presents many difficulties to fully resolve the mesoscale activity in the Algerian Basin because of the small eddy radii and the resolution limitations of altimetry data. The structures moving across the basin occasionally encounter

other structures and can deviate from their pathway, lose intensity or merge in a new stronger eddy (coalescence). These different behaviors are not always recognised by the algorithm, which can mistakenly assume the termination of an eddy and the formation of a new one. Additionally, the daily SLA data are the results of the interpolation of several satellites in different and complementary temporal ranges. As a consequence, any weak signal in vorticity could be lost. Furthermore, noise in the

SLA field or temporary distortions of the shapes of the eddies contribute to the errors in the tracking phase.

The most evident weakness of our algorithm concerns the presence of gaps in the detected tracks. Occasionally, some eddies bump into other features or into meanders of the current, losing vorticity and becoming undetectable by the W-based algorithm. The tracking method is not able to recognise these gaps and attributes a new ID to the newly-formed eddy, which typically occurred a few days after the termination of another one, in the same position. Furthermore, the spatio-temporal heterogeneity

of the altimetric tracks of satellites could introduce such lack of detection (Le Vu et al., 2017). As a result, the lifetime of the eddies could be incorrect and the tracks could be fragmented as well.

In order to solve this problem, we performed some improvements to the tracking methods by applying an eddy-continuity routine. For each vanishing eddy, the eddy-continuity algorithm examines all the structures which form in the following 7 days and computes the spatial distance and the difference in EKE between the new eddies and the vanished structure. If the distance

is less than $50\,\mathrm{Km}$ (mean eddy radius in the study area) and simultaneously if the difference in energy is less than the $30\%$ of the final EKE of the vanishing eddy, the algorithm associates the two structures, assigning to them the same ID. These threshold values are the results of sensitivity tests, obtained by the comparison of the tracks of several eddies with the maps of SLA.

The application of the continuity routine over 24 years leads to the decrease of the total number of eddies detected (from 8208 to 6543) with the consequent increase of the mean lifespan from 66 to 88 days.

To assess the strength of the eddy-continuity routine, we compared its output with the main parameters of a mesoscale eddy described in literature on the basis of in situ and satellite data. We focus on the eddy with coordinates $3.8°$ E and $38.3°$ N, described by Cotroneo et al. (2016), who indicates the formation of the eddy in June, $2014$, as supported by the application of the tracking methods of Mason et al. (2014). Figure 2 shows all the eddies of the domain on four different days from $23^{rd}$ August to $7^{th}$ September. The black dots indicate the W contours and thus the structures detected by the algorithm. Until

August $23^{rd}$, 2014 (top-left), the algorithm detected the eddy born on June $13^{rd}$. On August $24^{th}$ (top-right), the structure has interacted with another stronger eddy and lost vorticity. In the following 6 days the algorithm did not find any structure in that position. Thanks to the continuity routine, we found the same eddy 7 days later, on August $30^{th}$ (bottom-left). The same occurred after a few day, with a jump of 3 days. After the second re-identification the eddy became stronger in vorticity, amplitude and EKE, moved north-westward, distancing the eastern stronger structure (bottom-right). From here on out the

detection algorithm did not find any gap in the eddy track. The eddy terminates on $1^{st}$ January, 2015. Figure 3 shows the tracks of the eddy before and after the application of the eddy continuity algorithm. The red line indicates the correction to the original track (blue). In the first case the results suggest a lifetime of 72 days, while after the correction the lifetime is 203 days. Another example which confirms the improvement of the eddy continuity routine is presented through the analysis of the well-known structure described by Puillat et al. (2002), who observed an eddy with lifetime of about 3 years. With the

eddy detection and tracking routine we identify an anticyclonic eddy 410 days long (Figure 4). After this period, one day after

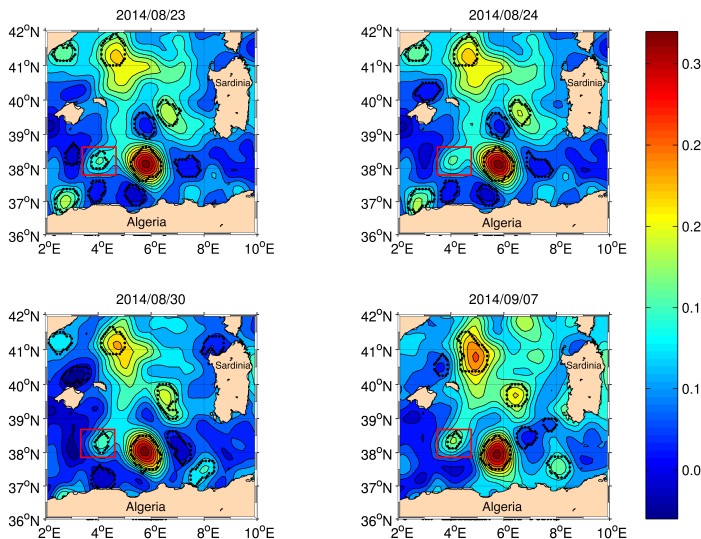

**Figure 2.** Maps of SLA (m) of four different days. The black dots indicate the Okubo Weiss closed contour and thus the structures detected by the algorithm. The red square indicates the eddy not correctly identified by the detection algorithm alone. In fact, on $24^{th}$ August, 2014, the structure probably became weaker and undetectable with this method. By running the continuity routine we join the tracks and obtain a single eddy track 203 days long. This structure have been described and tracked by Cotroneo et al. (2016) who computed the same pathway.

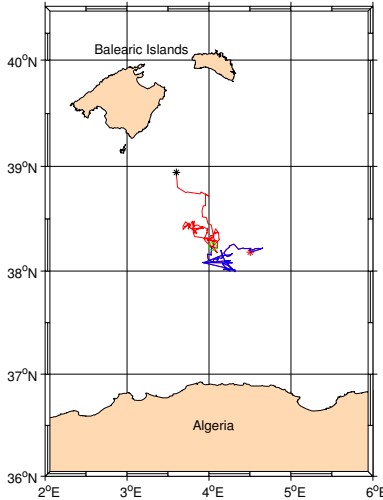

**Figure 3.** Tracks of the eddy described by Cotroneo et al. (2016) before/after the application of the eddy-continuity routine. Blue line shows the track without running the continuity routine (the red star indicates the formation point), while the red line (star) indicates the new part of the trajectory (termination). The green circle show the junction of the two tracks.

the termination, the routine finds another long-life eddy 295 days long. The eddy continuity routine joins the two structure into a single track, 706 days long. The eddy forms on December $25^{th}$, 1995 between $5°$ and $6°$ E at $37°$ N, moves around the basin and along the Eastern Algerian Gyre's pathway (EAG) (Testor et al., 2005b), and terminates in proximity to the Sardinia Channel, close to the Algerian coast on November $29^{th}$, 1997. So far the track proposed by Puillat et al. (2002) corresponds to ours. At the end of this period, near the coast, The SLA maps show the weakening of the structure and a splitting of the closed W contour. This area has been identified as a splitting zone also by Le Vu et al. (2017). The difference in the detection of the termination of this structure between the literature and our study is probably due to the different resolution of the SLA data and to the different altimeter missions used to compute the maps. In general, near the coastline, altimetry data are not reliable and it is hard to say if there the structure actually terminates and a new one forms or if it simply becomes weaker without terminating. Ideally, the properties of the eddies (such as the radius) are smaller during the formation phase, increases during

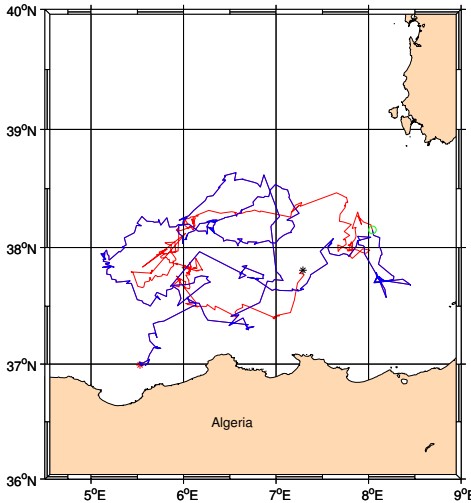

**Figure 4.** Tracks of the eddy described by Puillat et al. (2002) before/after the application of the eddy-continuity routine. Blue line shows the track without running the continuity routine (the red star indicates the formation point), while the red line (star) indicates the new part of the trajectory (termination). The green circle show the junction of the two tracks.

their life and decreases in the last phase, before the termination (Escudier et al., 2016b). This trend has been observed also for the kinetic energy (not shown), despite some oscillations around its mean value all during the life of the structure occur.

We computed the mean values of both the initial and final kinetic energies of all the short- and long-life eddies before and after the application of the continuity routine. In the first case we found some long-life eddies with an initial kinetic energy higher than twice the standard deviation. The application of the continuity routine solve this problem and consequently decreases the standard deviations of the initial and final kinetic energies.

In conclusion, we can affirm that the continuity routine we have implemented improves the results, even if it does not resolve all the problems of merging and splitting of the eddies. The latter topics, important for a more accurate description of the dynamical evolution of coherent eddies, have been recently tackled by Le Vu et al. (2017).

## 2.4 Methods limitations

The main advantage of the methods described in the previous sections (both physical and hybrid) is that they are automated and allow a large amount of data to be taken into consideration. Several years of satellite altimetry observations are thus used to investigate the mesoscale eddy seasonal and inter-annual variability. However, all methods mentioned above, when applied to altimetry data, take into account only the geostrophic velocities, while during their formation/vanishing phases the eddies are also subject to the ageostrophic components of velocity. Nevertheless, in order to track the eddies, the latter are considered as negligible.

The EKE of each eddy has been computed using the geostrophic velocities anomaly $u$ and $v$. Thus, the energy extrapolated from altimetry only describes geostrophic speed, whereas the "total kinetic energy" should include the ageostrophic components too and it is known that altimetry data can underestimate EKE by about half of its actual value (Pujol and Larnicol, 2005). However, in the study of the mesoscale circulation, the use of such data produces acceptable results.

Another issue with automated detection algorithms is the practical difficulty in defining an eddy boundary. Since eddies are fluid structures, time-dependent and continuously evolving without persistence, it is difficult to clearly demarcate their boundaries. There is some arbitrariness both in the definition of mesoscale eddies and in the method selected to define their borders (Chelton et al., 2011).

In some cases, the identification of the geometry and of the boundaries can be further complicated as small eddies arise from the splitting of a bigger structure, as happens to the eddy described by Cotroneo et al. (2016).

Every method based on the calculation of the W parameter requires a threshold value, which is critical in the identification of the eddies. There is no single optimal value for the global ocean and too high or too low a value can fail to detect small features or overestimate the size of the eddy. In the latter case, the eddies with an unrealistic area may encompass multiple vortices, sometimes with opposite polarities. A sensitivity test has been conducted varying the threshold value of W and our choice is discussed in Appendix A.

Furthermore, in order reduce this problem, by using the hybrid method, we first detect the closed loop of SLA around the extrema and then, within the SLA loop, the closed loop of W. Another problem of the physical methods is that the numerical computation of W is subject to noise in the SLA fields. In fact, the calculation of the second derivative of SLA fields amplifies the error (Chelton et al., 2011; Kurian et al., 2011). The application of the Hanning filter notably reduces the noise, but at the same time weakens the signal of some structures, making them difficult to be detected by the algorithm. The problem is solved by the application of the eddy-continuity routine, which fills the gaps in the tracks. It is also important to set a maximum radius to avoid mistakes in the eddy detection.

The association of regions of negative $W$ and regions embedded in SLA closed loops avoids the detection of spurious features due to noise and removes the ambiguities in multi-poles/elongated closed loops (Halo et al., 2014). Furthermore, structures with lifetime shorter than 7 days are not considered in the analysis.

## 3 Results

In our analysis we considered cyclones and anticyclones with lifetime greater than 7 days. The number of the cyclonic structures is greater than the number of the anticyclonic structures, but they have shorter lifespans and lower kinetic energy. The distribution of the anticyclones lifetimes suggests to study structures shorter and longer than 90 days separately, as 95% of them are short-life eddies and in particular, north of 39° N within the $97^{th}$ percentile the structure are shorter than 90 days. The case of the cyclones is different, as they have lifespans shorter than 90 days within the $98^{th}$ percentile and thus the analysis includes only cyclones shorter than 90 days.

Figure 5 shows the number of the cyclonic (A) and anticyclonic (B) eddy centres from 1993 to 2016 resampled in cells of $1/5°$ x $1/5°$. Three areas with the greater concentration of eddies coincide for cyclones and anticyclones. We found the highest number of centres (more than 200 per sampling unit) in the Sardinia Channel (37.5° N, 8° E), known in literature as an eddy formation area (Escudier et al., 2016a). At the same latitude, at about 6° E, we detect a second maximum. The third maximum is located in the northern region of the basin, at about 40° N, corresponding to the path of the WMMC, in proximity of the NBF. The cyclones also present a high concentration (up to 250 eddies) around the Greenwich meridian.

It is important to underline that south-west of the Sardinian slope, close to the coast, we found less than 50 eddy centres. This area is characterised by upwelling phenomena (Olita et al., 2013) and by a quasi-permanent cyclonic circulation: the Southerly Sardinia Current (SSC) (Pinardi et al., 2015).

Our results are comparable with those of Escudier et al. (2016b) and Le Vu et al. (2017), showing similar high density areas in the Algerian Basin. It is worth to note that a seesaw pattern in the eddy number distribution is visible also in our maps and it is probably due to the sampling tracks of satellite altimeters.

Figure 6 shows the absolute value of the averaged relative vorticity, normalised with respect to the Coriolis parameter ($f$), $\xi/f$, for all the cyclonic (A) and anticyclonic (B) structures detected over the full period. The cyclonic structures present lower vorticity (in absolute value) and a spatial distribution more homogeneous than anticyclones. The values are around $0.04 - 0.09$ (greater close to the coast) for cyclonic features and around $-0.10$-$-0.15$ for the anticyclonic ones.

The area where we found the maximum number of centres, south of 39° N, corresponds to the area with higher values of vorticity, as it would be expected. The distribution of the averaged radii and amplitudes present a zonation similar to the above seen variables (not shown, while just EKE is shown in Figure 18).

The distributions in Figures 5 and 6 further suggest that the basin can be divided into two sub areas, north of and south of 39° N.

Figure 7 shows the time series of monthly mean EKE (see Eq. A5 in Appendix A) of the cyclonic (dashed lines) and anticyclonic (solid lines) eddies from 39° N to 42° N (blue) and from 37° N to 39° N (red). In particular, we computed the monthly mean of the daily energy of each centre.

The values for the anticyclonic northern eddies fall in the range from 50 to 100 $cm^2/s^2$ with peaks in the years $1998 - 1999$, 2006, 2010 and 2013. The energy of the anticyclonic southern structures shows the highest inter-annual variability with values about two or three times greater than the northern ones. The mean is centred on 200 $cm^2/s^2$ with peaks greater than 250

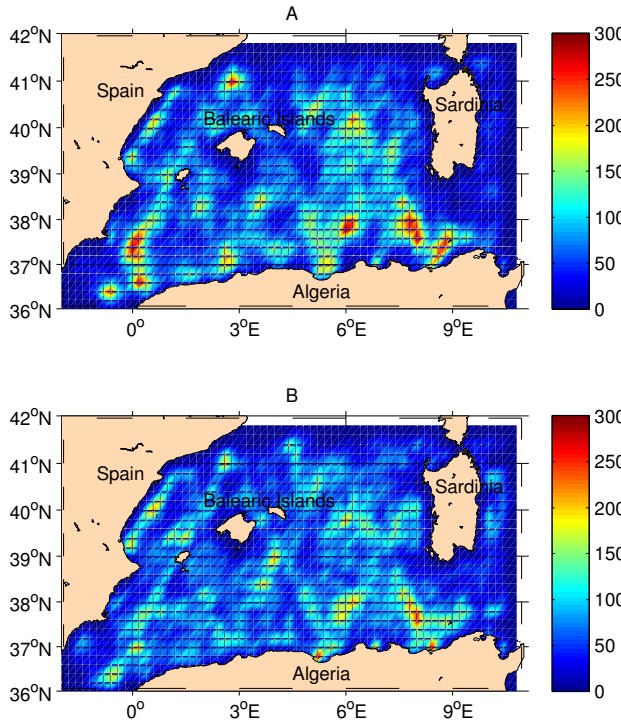

**Figure 5.** Number of the cyclonic (A) and anticyclonic (B) eddy centres from January 1993 to December 2016 with a grid resolution of $1/5°$. The highest density is in proximity of the Sardinia Channel.

$cm^2/s^2$. The first peak, centred in 1997, has been detected also from Pujol and Larnicol (2005). The other maxima have been found in the periods November, 2004 - August, 2006, May, 2008 - May, 2010, December, 2012 - December, 2014 and November, 2016. Most of them have a duration greater than one year.

It is important to note that the energy of the cyclonic structures is generally lower than that of the anticyclonic ones, which thus have a stronger impact on the mesoscale circulation of the Algerian Basin.

Hereafter we will focus the attention on the anticyclonic structures, although we will present some results about cyclones. To evaluate the relationship between the lifetime and the properties of the eddies we calculated the mean kinetic energy (MEKE) and the mean radius of all the eddies with lifetime within a specific temporal range. The lifetime has been divided into ranges, of approximately 30 days each, from 7 up to 180 and 240 respectively for cyclones and anticyclones. The last range for both kinds of structures includes structures longer than 6 and 8 months respectively. Figures 8 and 9 - Panels A show the averaged values and the standard deviation of the MEKE of the eddies for each lifetime range. For both the cyclonic and the anticyclonic structures the relationship increases almost monotonically and is characterized by high standard deviation. Cyclones have mean kinetic energy about three times smaller than anticyclones.

The same study has been conducted for the mean radius of each eddy (Figures 8 and 9 - Panels B). Similarly, the structures

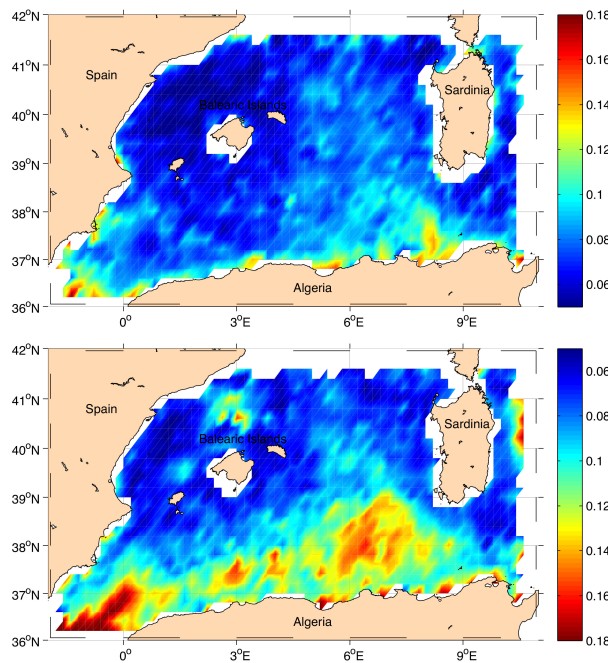

**Figure 6.** Absolute value of the mean relative vorticity normalised with respect to the Coriolis parameter ($f$), $\xi/f$, of the cyclonic (A) and anticyclonic (B) structures from January, 1993 to December, 2016, with a grid resolution of $1/5°$. The largest vorticity (in absolute value) is associated to anticyclones south of $39°$ N

with a short lifetime have a small radius (on average 30 Km for the first step). The mean radius reaches values around 40 Km for eddies longer than 90 days. It increases for eddies longer than 240 days, reaching 43 Km. In general, the radii of cyclones are comparable to those of anticyclones and also reach values around 47 Km. To identify the areas of formation, convergence, detaching and vanishing of the eddies we superimposed a regular grid on the domain. The 9 sectors are labeled from A to I (Figure 11). Each is $2°$ of latitude and $2°$ of longitude, from $1.5°$ W to $8.5°$ E and from $36°$ N to $41°$ N. We posit that the mesoscale structures have different pathways and lifetime depending on their area of formation and kinetic energy. Sector A is shown in the interest of completeness but it is not examined in any detail because it falls outside our area of study.

The statistical analysis has been conducted on cyclones with lifespan up to 90 days (98% of the cyclonic structures) and on anticyclones both shorter and longer than 90 days.

## Cyclonic eddies

The histograms in Figure 10 show the number of cyclonic eddies formed (red bars) and vanished (blue bars) in each sector. The yellow bars indicate the number of structures which have both formed and vanished in each sector. The difference between the red and the yellow bars represents the number of eddies coming from other parts of the basin and vanishing in a given sector.

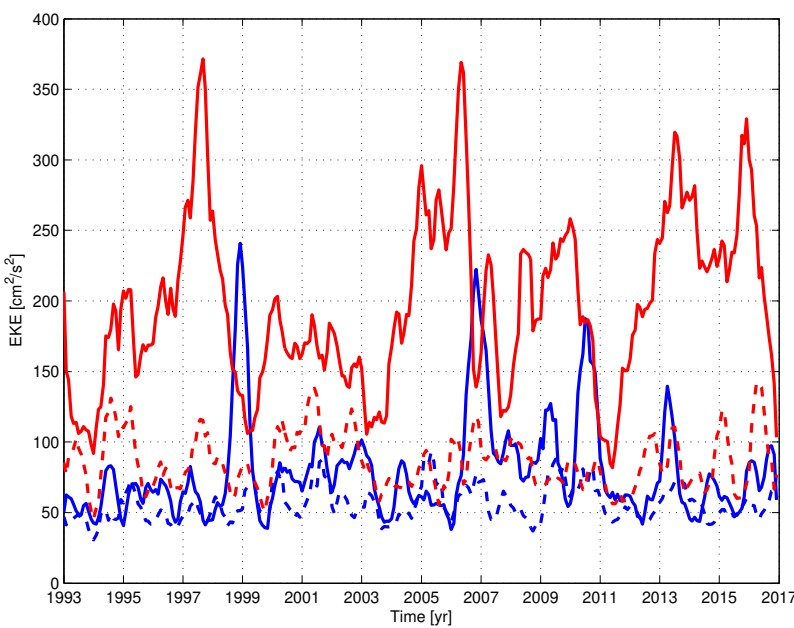

**Figure 7.** Time series of the monthly mean EKE (mmEKE) from 1993 to 2016. Red lines refer to the mmEKE of anticyclonic (solid) and cyclonic (dashed) eddies in the southern part of the basin (37°-39° N), while blue lines refer to the mmEKE of anticyclonic (solid) and cyclonic (dashed) eddies in the northern part of the basin (39°-42° N).

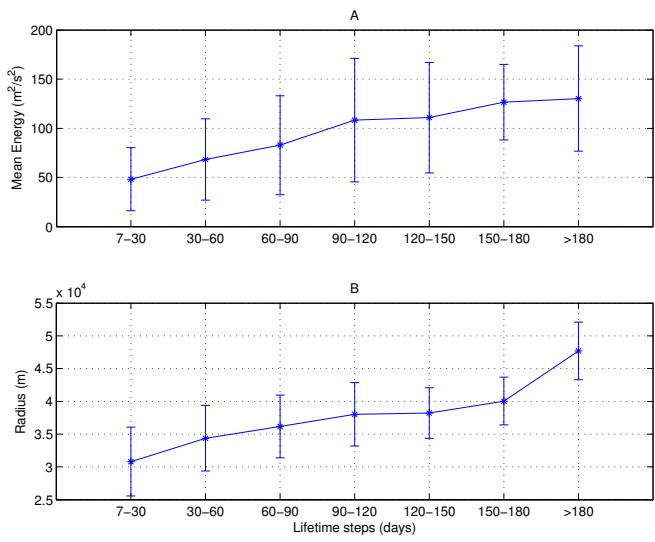

**Figure 8.** Relation between MEKE (A), radius (B) and lifetime of cyclonic structures. Averages have been computed on the basis of MEKEs and mean radii of the cyclonic eddies in the basin.

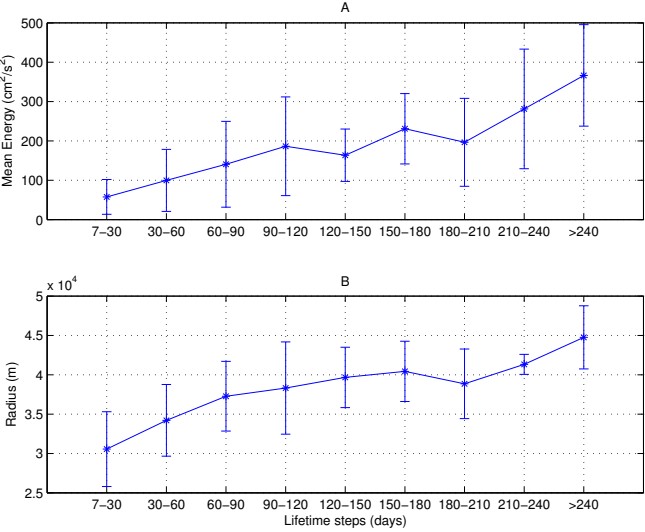

**Figure 9.** Relation between MEKE (A), radius (B) and lifetime of anticyclonic structures. Averages have been computed on the basis of MEKEs and mean radii of the anticyclonic eddies in the basin.

It could be an indirect indication for zone of convergence.

The number of formations and terminations are slightly higher in the south-western sectors, along the AC (F, G). Figure 11 shows the track of the structures formed in sectors C and G, which correspond to the maxima of Figure 10 along the transect between Balearic Island and Sardinia (northern part of the basin) and along the Algerian coast (southern part of the basin) respectively. The cyclonic structures do not have specific pathways and they tend to remain close to the formation area. The eddies formed in sector G are transported eastward by the AC and reach sector H, which is a termination area.

**Short-life anticyclonic eddies (shorter than 90 days)**

The same analysis has been computed for short life structures (Figure 12). The higher number of formations is concentrated in sectors A and C (more than 300 eddies), and more generally the northern part of the basin (A-D) hosts more formations of anticyclonic short-life eddies (1028) than the 828 of the southern part (F-I). Furthermore, the yellow bars show that almost all the eddies formed in each of the nine sectors vanished in their sector of formation (around 80% in both southern and northern parts). Figure 13 shows the tracks of the eddies formed in sectors C, F and I chosen as they are the areas with the higher number of formations respectively in the northern part (along the transect between Balearic Islands and Sardinia) and in the southern part of the basin. Short-life eddies do not move around the basin, but rather they remain close to their area of formation. In fact, despite their mean translational velocity of $4\ Km/day$ and an average lifetime of $30 - 40$ days, they tend to follow a random pathway and do not move far from where they originated. In general, just a few eddies, usually the more energetic ones, move to another sector. Some of the structures formed in F follow the pathway of the AC and move along the coast, reaching the next

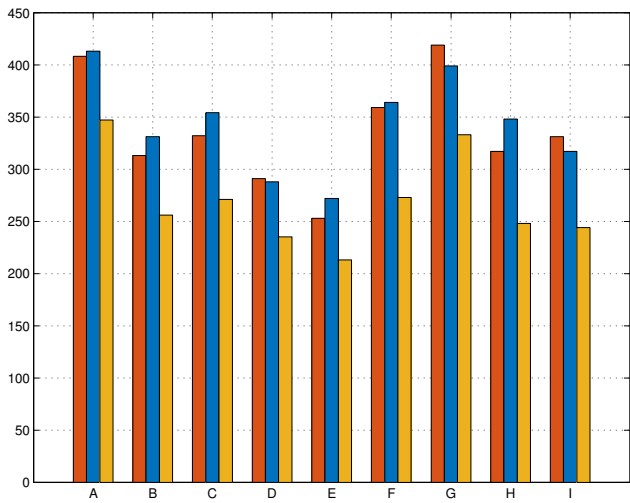

**Figure 10.** Number of cyclonic eddies formed (red bars) and terminated (blue bars) in each sector. The yellow bars indicate the structures formed and terminated in the same sector.

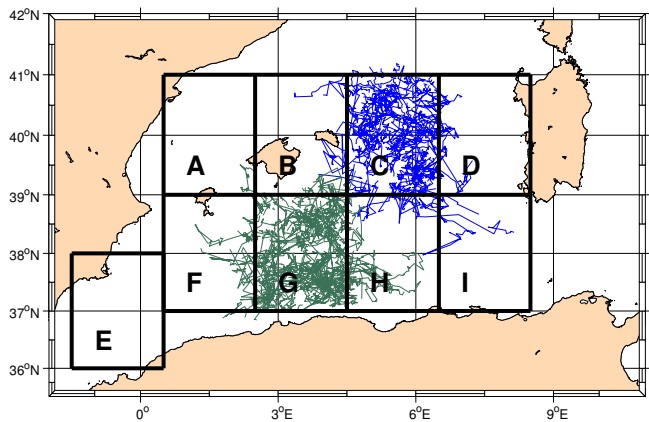

**Figure 11.** Tracks of the cyclonic eddies formed in sectors C (blue) and G (green)

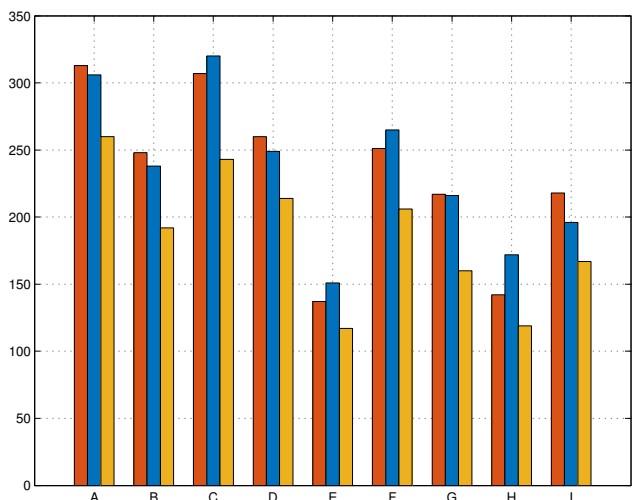

**Figure 12.** Number of short-life anticyclonic eddies formed (red bars) and terminated blue bars) in each sector. The yellow bars indicate the structures formed and terminated in the same sector.

eastern sector. Almost all the structures born in sector I do not pass across the Sardinia Channel but vanish west of $9°$ E. Just the $3\%$ of the eddies born east of $8°$ E can pass through the strait and vanish east of $9°$ E. The structures formed in the areas

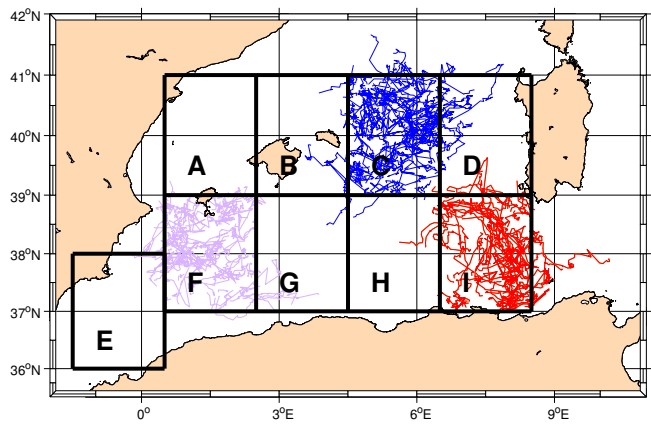

**Figure 13.** Tracks of the short-life eddies formed in sectors C (blue), F (lilac) and I (red).

B, C and D typically do not cross $39°$ N, except for a few cases. These structures have been detected in the past by Fuda et al. (2000), who labelled them Frontal Eddies (FEs), considering only structure with lifetime up to $15$ days. We use the name FEs to indicate short- or long-life cyclonic and anticyclonic eddies, formed along the NBF. We hypothesize they are caused by the instabilities of the NBF.

**Long-life anticyclonic eddies (longer than 90 days)**

The same analyses have been performed for anticyclonic eddies with lifetime greater than 90 days (Figure 14). The biggest number formed is in the Sardinia Channel, between $6.5°$ E and $8.5°$ E (sector I, 28 eddies). The AEs move cyclonically around the basin, crossing several sectors, typically with their centre remaining south of $39°$ N, sometimes completing an entire loop (Figure 15). They often vanish in sector H, where we found the biggest number of terminations (28). $54\%$ of the FEs follow the same cyclonic loop as well (Figure 16). In sector H the difference between the number of terminations of eddies coming from other areas and eddies born there is very high (22): it suggests the area to be a convergence zone for the long-life anticyclonic eddies, as also highlighted by the large number of features observed in this sector (Figure 5). As this area lies between two cyclonic gyres (Figure 18) the eddies formed west of $3.5°$ (sector G) move mainly westward, following the western cyclonic gyre (Western Algerian Gyre - WAG (Testor et al., 2005b)), while the eastern structures of sector G move eastward along the AC.

It is important to underline that all the long-life eddies formed in sector I do not pass the Sardinia Channel, but deviate northwards and follow the eastern cyclonic loop.

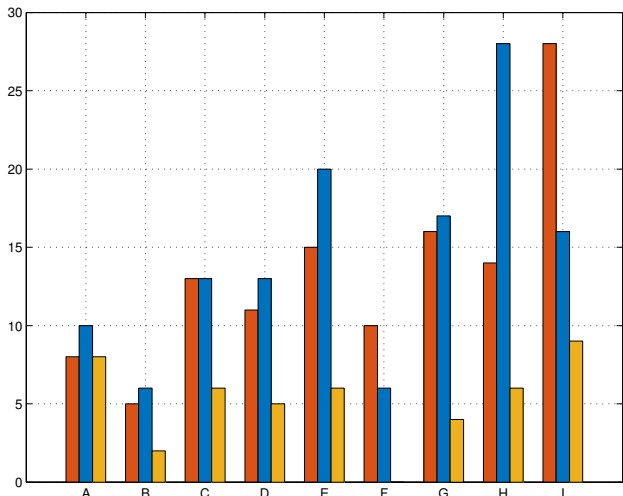

**Figure 14.** Number of long-life anticyclonic eddies formed (red bars) and terminated (blue bars) in each sector. The yellow bars indicate the structures formed and terminated in the same sector.

**Long-life eddies and EKE peaks**

In Figure 7 we showed five energy peaks longer than one year in the time series of anticyclones of the southern part of the basin. Additionally, Figure 9 suggests the long-life eddies to be the most energetic structures, with a mean energy three times greater than cyclones and short lived anticyclones. The five peaks of the southern part of the basin actually correspond to the presence of structures with lifespan greater than $450$ days. We found 6 eddies (Figure 17) in the years $1995 - 1997$ (Panel A),

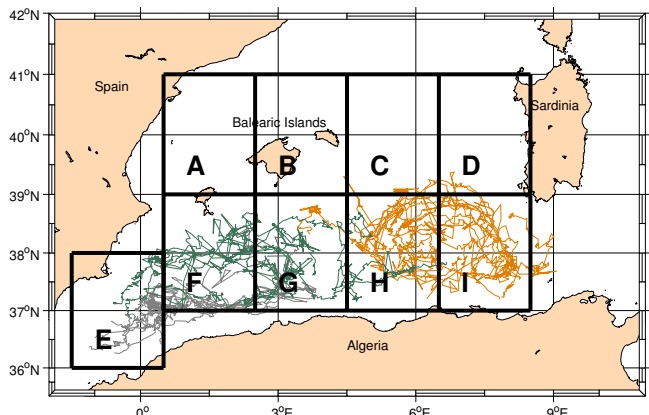

**Figure 15.** Tracks of the anticyclonic AEs born in sectors E (grey), G (green) and I (orange). The eddies formed west of $3.5°$ (sector G) move mainly westward, following the western cyclonic gyre (Figure 18), while structures formed east of $3.5°$ move eastward along the Algerian Current.

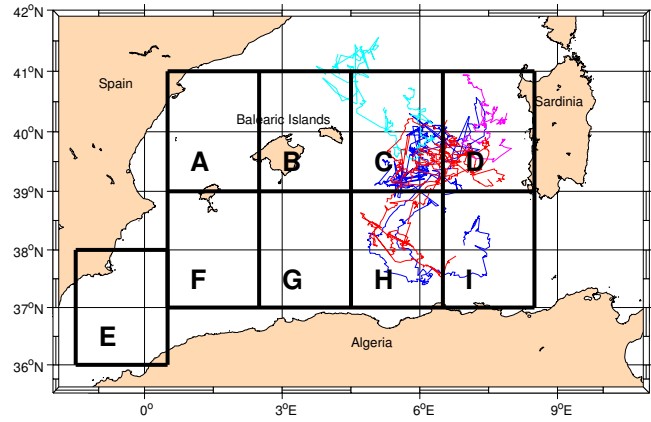

**Figure 16.** Tracks of the anticyclonic FEs born in sectors C (blue and cyan) and D (red and magenta). Blue and red lined indicates the eddies which follow the cyclonic loop and move southward, while red and magenta lines indicates the structures which move northward.

2004 − 2007 (Panel B), 2009 − 2010 (Panel C), 2012 − 2014 (Panel D) and 2015 − 2016 (Panel E).

The first peak in Figure 7, corresponding to the highest values of mmEKE, is related to the longest track which starts on December, $25^{th}$, 1995 and terminates on November $19^{th}$, 1997 (Panel A). It is the well-known long-life structure 96/1 studied by Puillat et al. (2002). The differences in the tracks between literature and our work have been discussed in Section 2.3. The second high peak, in the period from 2004 to 2007, is associated with two long-life eddies (Panel B). One of the two long-life structures moves northward (green line) and is probably responsible for the peak in the northern part of the basin in the year 2006. The same happens in 2009 − 2010, when we observe a peak in the southern part followed by another peak in the northern part (Figure 7). Also in this case the eddy has been involved by the Algerian Gyre and after having passed the 39° N it moves northward transferring energy to the northern part of the basin. The last two peaks are associated to two long-life eddies forming respectively in 2012 and 2015 (Panels D and E).

As suggested by previous analyses, all the long-life eddies spend the most of their life south of 39° N, except for two structures which terminate in the northern basin. The area where the features can cross 39° N is in the middle of the basin (between 5° E and 7° E) and is characterised by large (negative) mean relative vorticity (Figure 6).

The analysis of correlation and lag correlation between the two time series of the northern and southern anticyclonic structures did not show significant results and thus we suppose that long-life eddies can transfer an amount of energy from one area to another one. Nevertheless, we did not find any particular track corresponding to the first peak in the northern part of the basin in 1999, that will be further investigated in future.

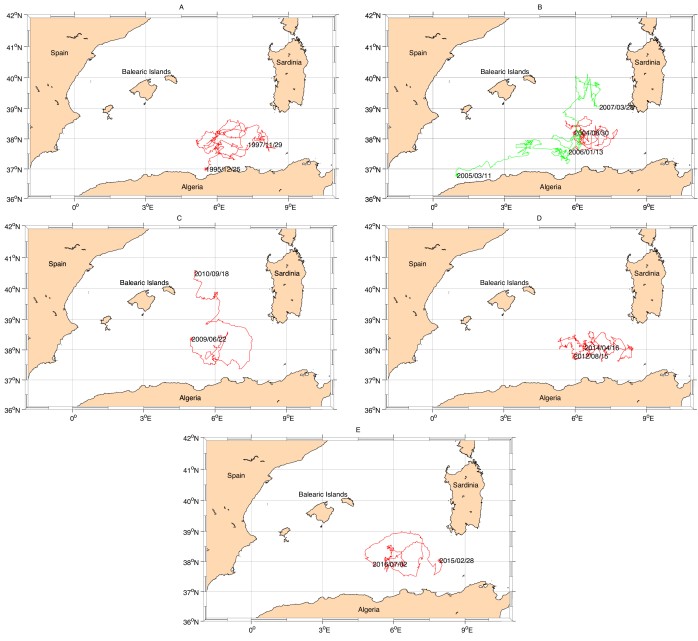

**Figure 17.** Tracks of structures with lifespan greater than 450 days. The dates indicate the formation and the termination of each eddy.

**Seasonality**

Thanks to the availability of a large amount of data, it has been possible to evaluate the seasonal variability of the eddy formation in the basin (See Table 1).

The distribution for cyclonic and anticyclonic structures with lifetime shorter than 90 days is almost homogeneous, with a peak in winter, i.e. from October to March. On the contrary, the long-life anticyclonic eddies form preferably in spring, i.e from April to June, according to the maximum transport through the Strait of Gibraltar which occurs in March (Beranger et al., 2005). The seasonality of the long-life structures is thus shifted in time around $3 - 5$ months with respect to the cyclonic and anticyclonic features with lifespans shorter than 90 days.

| | Winter | Spring | Summer | Fall |
|---|---|---|---|---|
| **Cyclonic eddies** | 1132 | 1042 | 1011 | 964 |
| **Anticyclonic short-life eddies** | 906 | 786 | 641 | 814 |
| **Anticyclonic long-life eddies** | 32 | 54 | 35 | 18 |

**Table 1.** The Table shows the number of formations of cyclonic and anticyclonic eddies in each season for the period 1993 to 2016. Winter includes the months from January to March, spring from April to June, summer from July to September and fall from October to December.

**Eddy translational velocity**

In order to find the main translation direction of the eddies, we divided the basin into a regular grid of $1/5°$ and we computed the average values of the components $u$ and $v$ of the translational velocity (Figure 18). The mean translational speed of all the eddies in the basin is 3.2 cm/s. The arrows in Figure 18 indicate the mean translational velocity for cyclonic (panel A) and anticyclonic structures (Panel B). The colour indicates the mean kinetic energy. Cyclones do not present any preferred direction and, as expected, are characterized by lower energies than the anticyclonic structures. On the contrary, the latter highlight the present of the two Algerian Gyres (WAG and EAG). The WAG, between $0°$ and $3.5°$ E, is responsible of the westward movement of the anticyclonic eddies formed in sector G, west of $3.5°$. The map also confirms that on average the anticyclonic eddies follow the AC till $8°$ E. The highest speeds (up to 9 cm/s) are reached in proximity to the Sardinia Channel, along the AC and offshore of the current, where the anticyclonic structures detach from the coast and deviate northward entering in a cyclonic loop (EAG), south of $39°$ N. The eastern cyclonic loop is also characterized by high speeds and the highest kinetic energy (up to $360\ cm^2/s^2$). Lower speeds are found in the middle of the basin, where the eddies interact with the AC and other structures. Furthermore, the area in correspondance with the EAG hosts a large number of centres (Figure 5), and a maximum of (negative) relative vorticity (Figure 6), making this region of particular interest from the dynamical point of view. In the northern part of the basin, north of $39°$ N, the mean translational velocities are less than in the southern part and do not present a predominant direction.

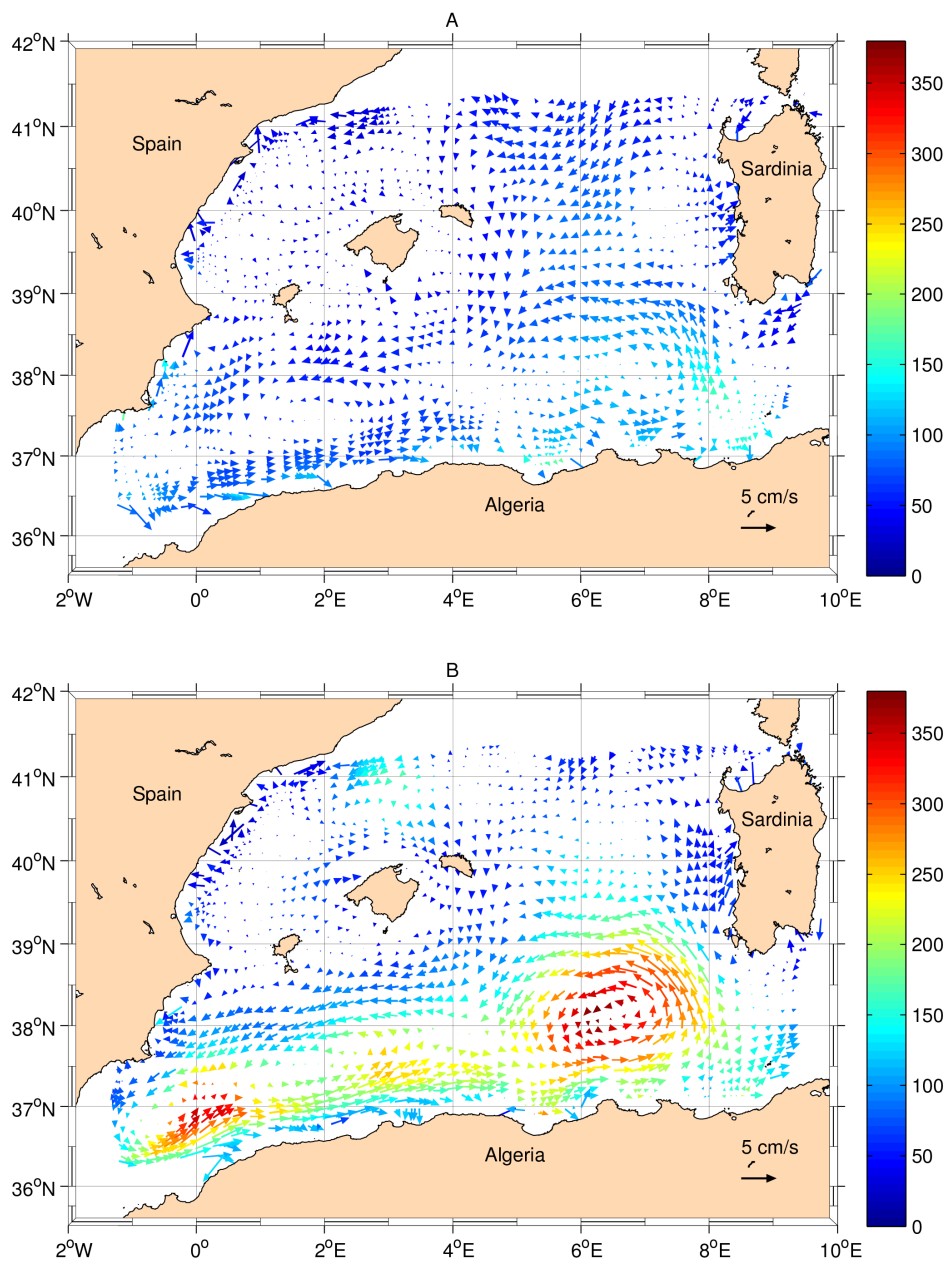

**Figure 18.** Average velocities in cm/s of the cyclonic (A) and anticyclonic (B) eddies in boxes of $1/5°$. The colour indicates the average daily EKEs ($cm^2/s^2$) in the same boxes

## 4 Summary and conclusions

The automated method for eddy detection and tracking, applied over 24 years of merged altimetry SLA maps, allows us to characterise the variability of mesoscale structures and, in general, describe mesoscale circulation, which is strongly influenced by the pathways of the eddies within the basin.

As mentioned in Section 2, the hybrid method presents a number of advantages reducing the disadvantages of other methods. It is important to note that the tracking method is based on the definition of a general distance (Eq. 1), which in turn defines the evolution of each eddy over time. In a basin as small as the Algerian Basin, eddies frequently encounter other eddies and deviate from their paths or merge into new, larger structures (coalescence). A significant limitation of the algorithm is its inability to recognise weak signals in the eddy vorticity (causing gaps in the continuous tracking, here called *jumps*), and as a

consequence, the misidentification of old eddies as new ones. This limitation could affects both the definition of a pathway and the estimate of their lifetime and number. To overcome this limitation, we implemented an eddy-continuity routine, which detects if, in the 7 days following a death, a new eddy arises with close values of EKE and in a close position with respect to the previously identified eddy. The comparison with existing literature (Puillat et al., 2002; Escudier et al., 2016a; Cotroneo et al., 2016) and some sensitivity tests (not shown) have demonstrated the reliability of the detection/tracking algorithm and

the improvement given by the implementation of the eddy-continuity routine.

After estimating the lifetime of all the eddies, we decided to ignore structures with lifetime shorter than 7 days, as the algorithm may introduce spurious short-lived structures. We performed a eulerian statistic on cyclonic eddies shorter than 90 days (98%) and on anticyclonic eddies both shorter and longer than 90 days separately, in order to study their spatial distribution. We found that the cyclonic eddies have in general shorter lifespans and lower energy than the anticyclonic structures. They do not have

specific pathways and do not move far from the area of formation. Over 24 years of analysis, we estimated the total amount of kinetic energy of the cyclones to be some 30% lower than those of anticyclones. Their characteristics are similar to those of the anticyclonic short-life structures, which mostly form in the northern part of the basin (58% north of 39° N) and are less energetic. They barely move within the basin and terminate close to their area of formation. The remaining 42% of short-life eddies, mostly found south of 39° N, have higher kinetic energy.

This result complements that of anticyclonic long-life eddies, which typically form south of 39° N (63%). The main area of formation is located from 6.5° E and 8.5° E and from 37° N to 39° N (sector I), in proximity to the Sardinia Channel. Long-life eddies probably absorb energy from the Algerian Current and move cyclonically within the basin, following the Eastern Algerian Gyre, without moving north of 39° N. Sector I can be considered not only as a formation zone, but also as a detaching area, where coastal eddies become open-sea eddies. The remaining 37% of long-life eddies form north of 39° N (sectors A-D).

The most energetic ones move southward and join the eastern cyclonic loop (Eastern Algerian Gyre).

We detected an area of convergence corresponding to sector H, from 4.5° E to 6.5° E, between the Western and the Eastern Algerian Gyres. Long-life features forming in sector I or coming from the northern basin (sectors C and D, from 4.5° to 8.5° E and from 39° N to 41° N) terminate there. Also, some of the short-life AEs formed in sector G (from 2.5° E to 4.5° E and from 37° N to 39° N) following the Algerian Current die in sector H, as do some short-life structures coming from sectors C and D.

These short-life structures are known in literature not to become open sea eddies (Obaton et al., 2000; Millot et al., 1997). In this area the Algerian Current spreads seaward for months (Benzhora and Millot, 1995a; Millot et al., 1997) probably due to the presence of the Eastern Algerian Gyre.

The monthly mean kinetic energy of the structures shows that the southern part of the basin is about three times more energetic than the northern one (up to $300\ cm^2/s^2$) and is subject to annual periodicity (not shown). This is in accordance with the study of Pujol and Larnicol (2005) which also highlights that the seasonal EKE variations are mainly concentrated southwest of Sardinia. The five peaks of energy found in the EKE time series in the northern basin correspond to the eddies with lifetime greater than 450 days. It confirms the proportionality between lifetime and eddy kinetic energy. The maximum centred in the year 1997 is also present in the EKE computed at the basin scale by Pujol and Larnicol (2005).

By examining the properties of each eddy, we found that the longer the lifetime of the structures, the larger their radius and the higher their kinetic energy.

In literature, southern eddies formed along the Algerian Current are commonly referred to as Algerian Eddies (AEs). In order to distinguish the eddies formed in the two parts of the basin, we labelled the northern structures Frontal Eddies (FEs) according to Fuda et al. (2000), as we suppose that they are caused by the instabilities of the NBF. Northern and southern structures differ on the basis of lifetime, pathway and other properties such as energy and dimension.

The complementarity between anticyclonic short- and long-life structures has also been observed in the study of seasonality. Short-life anticyclonic eddies form mostly in autumn and winter, while anticyclonic long-life eddies typically form in spring in the southern basin, when the AC is at its maximum transport (Beranger et al., 2005). It was observed that the transport through the Gibraltar Strait is the physical parameter which supports the eddy formation mechanism (Obaton et al., 2000), strengthening the density gradient and facilitating baroclinic instabilities.

The eastern cyclonic loop of the eddies in the southern part of the basin corresponds to the high-energy and high-density pattern, and is partly superimposed on the Eastern Algerian Gyre, flowing in the LIW layer (Testor et al., 2005b; Escudier et al., 2016a). The along-slope propagation of southern AEs stops in proximity to the Sardinia Channel, where the structures probably interact with the bathymetry and with a salinity (and density) barrier, which may contribute to blocking the propagation of eddies. We suppose the structures remain there until they absorb from the Algerian Current an amount of energy sufficient to detach from the coast and move northward.

The anticyclonic eddies in the Algerian Basin have different properties depending on their formation area. The northern anticyclonic FEs are mostly short lived and less energetic, with smaller relative vorticity (absolute value) and translational velocity than the southern AEs and without a well defined pathway. These structures do not seem to be related to AEs and to the Algerian Current; they are more probably caused by other currents, such as the North Balearic Front, or by wind forcing (Le Vu et al., 2017). The interaction between these two kind of structures is represented by a few anticyclonic FEs, which deviate southward and join the cyclonic loop, in correspondence with the Eastern Algerian Gyre's pathway, and by a limited number of anticyclonic AEs that pass the $39°$ N in the middle of the basin and move northward, bringing energy to the northern part of the basin.

In accordance with Escudier et al. (2016a) and Le Vu et al. (2017) the long-life anticyclonic eddies form preferably in the

southern part of the basin, from $1.5°$ W to $0.5°$ E, from $2.5°$ E to $4.5°$ E and from $6.5°$ E to $8.5°$ E (sectors E, G and I); they move along the Algerian Current and detach from the coast to follow the two cyclonic gyres (Western and Easter Algerian Gyres). In particular, sector G hosts eddies deviating westwards and eddies deviating eastwards. The southern short-life anticyclonic eddies form preferably in sector F, from $0.5°$ E to $2.5°$ E, where also Escudier et al. (2016a) identified a weaker area of eddy formation. The area between the two gyres (sector H) correspond to a convergence zone, where we found several eddies interacting each other and with the main current.

These results may have implications on the dynamic of the whole basin and in particular on the estimation of the transport of physical and biological properties. The study of the trajectories of long-life anticyclonic features is important to better investigate the circulation of the entire basin and thus of the Western Mediterranean Sea. On the other hand, cyclones and in general short-life anticyclones have probably less influence over the circulation, but are important in the study of local phenomena, especially linked to biological processes. A future proposal is to investigate the vertical extent of some case-study structures with a multi platform approach, i.e. using in situ data, drifters and gliders combined with these results.

## Appendix A

### A1    Detection algorithm's main steps

The hybrid eddy detection method is based on the computation of the Okubo Weiss parameter within the closed SLA contours around the extrema (maxima and minima). The main steps of the algorithm are summarized in the following list:

- Selection of the spatial and temporal range to analyse.

- Selection of the interval between the SLA contours, which should approximate the precision of the altimetry. We choose $0.02$ m.

- An upper threshold for Okubo-Weiss parameter detection is fixed at $0$.

- A minimum value for SLA is fixed at $0$.

- Selection of the number of Hanning filter iterations on the Okubo-Weiss parameter. We choose $2$ passes.

- The algorithm finds the local extrema in SLA.

- The algorithm computes the geostrophic rotational velocities, the $W$ parameter, the vorticity and the Eddy Kinetic Energy (EKE).

- If the $W$ parameter is negative within a closed loop of SLA, the algorithm computes the number of grid points inside the eddy. If this number is greater than $4$, the detected structure can be considered a mesoscale eddy.

- The algorithm checks if there are other extrema in this eddy, with negative W. In this case the algorithm detects more separated structures.

- The code extrapolates all the properties of the eddy such as position, time, area, surface kinetic energy, vorticity, equivalent radius (computed from the area), maximum, minimum and mean sea surface height (SLA), amplitude, rotational speed and finally zonal and meridional propagation velocities.

- After the detection step, a routine selects the eddies with radius included in the range $0 - 300$ Km and amplitude greater than zero m (tunable parameters). Results from this step will be used later by the lagrangian routines in order to track the eddies.

## A2 Okubo-Weiss parameter threshold

The choice of the threshold value of the Okubo-Weiss parameter ($W_0$) has been largely discussed in literature (Sadarjoen and Post, 2000; Le Vu et al., 2017). Isern-Fontanet et al. (2004) suggest the value $W_0 = -0.2\sigma_0$, where $\sigma_0$ is the standard deviation in the Mediterranean Sea (around $10^{-11}$), while Chelton et al. (2007) propose a fixed value $W_0 = -2.1 * 10^{-12}$. During the first phase of eddy detection we conducted a sensitivity test over the entire period in the whole domain with $W_0 = 0$ and $W_0 = -2.1 * 10^{-12}$ (according to Chelton et al. (2007) and very close to the values suggested by Isern-Fontanet et al. (2004)). The results show that there is a decrease of $0.4\%$ in the number of anticyclones and an increase of $0.2\%$ in the number of the cyclones detected in the area of study using the latter value. The variation of the mean radius is around $3\%$ (smaller with $W_0 = -2.1 * 10^{-12}$) for both cyclones and anticyclones. The energy variation, using $W_0 = -2.1 * 10^{-12}$, is $0.4\%$ smaller for anticyclonic structures and $0.2\%$ smaller for the cyclonic ones. Despite these minor discrepancies in the mean values, some differences arise when looking at the specific properties of some well-known structures. In particular, we observed that in the case of $W_0 = 0$ the detection method (and consequently the tracking method) identifies a structure characterised by a lifetime of 610 days formed in the southern part of the basin. The same structure is unfortunately not well detected setting $W_0 = -2. * 10^{-12}$. In fact, with the latter value, the same structure has a lifespan of 178 days due to the misleading detection of three closed Okubo-Weiss contours (trifurcation). The comparison with the daily SLA maps suggests that the choice of $W_0 = 0$ in this case is better suited. Furthermore, setting $W_0 = -2.1 * 10^{-12}$ the algorithms tracked a structure 466 days long in the northern part of the basin, which appears shorter (296 days) with $W_0 = 0$. In this case, the analysis of daily SLA maps suggests that the use of $W_0 = 0$ is preferable.

It is important to note that the peaks of energy, which find correspondence in the tracks longer than 450 days, would not have a corresponding track in 2013 setting $W_0 = -2.1 * 10^{-12}$. In general, our idea is that the smaller value of $W_0$ ($W_0 = -2.1 * 10^{-12}$) highlights the daily presence of weak and small structures that could create "noise", and occasionally influence the detection of longer and stronger features.

Neither the relationship between lifetime and the mean kinetic energy and the mean radius nor their standard deviation differ significantly between the two different values of $W_0$.

## A3 Derived variables

In this section are described the principal variables used in the paper. Once the structures are identified by the detection algorithm, the kinetic energy of each point of the eddy is calculated starting from the geostrophic velocities. Throughout we will refer to the kinetic energy normalized by mass.

The total areal kinetic energy (*AEKE*) is defined as follows:

$$AEKE = \sum_{k=1}^{N} EKE_k ds_k \tag{A1}$$

where $EKE_k$ ($cm^2/s^2$) and $ds_k$ ($cm^2$) are respectively the kinetic energy and the unit of area for each point of the grid inside the eddy. $N$ is the number of unit of area inside the eddy.

The area of the eddy S ($cm^2$) is:

$$S = \sum_{k=1}^{N} ds_k. \tag{A2}$$

The *EKE* ($cm^2/s^2$) is the energy of each eddy centre, calculated as:

$$EKE = \frac{AEKE}{S}, \tag{A3}$$

The total daily EKE ($tdEKE$) is the sum of the EKEs of all the eddy centres for each day:

$$tdEKE = \sum_{i=1}^{neddies} EKE_i \tag{A4}$$

where $neddies$ is the number of the eddies of each day. It is measured in $cm^2/s^2$.

The monthly mean EKE ($cm^2/s^2$) is calculated as:

$$mmEKE = \frac{1}{T} \sum_{d=1}^{T} tdEKE_d \tag{A5}$$

where $T$ is the number of the days for each month.

We refer to the mean kinetic energy ($MEKE$) of each eddy with the formula:

$$MEKE = \frac{\sum_{d=1}^{lt} EKE_d}{lt}, \tag{A6}$$

where the $lt$ is the lifetime in days and the MEKE in $cm^2/s^2$.

## A4 Regional values adopted

The regional values adopted in the tracking algorithm are listed below:

- $L_0 = 80$ Km is the typical eddy distance ;

- $R_0 = 70$ Km is the typical eddy radius scale;

- $\xi_0 = 10^{-5}\ s^{-1}$ is the typical eddy vorticity scale;

- $Z_0 = 0.1$ m is the typical eddy mean SLA variations;

- $A_0 = 0.1$ m is the typical eddy SLA amplitudes variation.

*Acknowledgements.* This paper is supported by the and co-financed by the European Regional Development Found, by Italian Flagship Project RITMARE founded by the Italian Ministry for Research – MIUR (NRP 2011-2013, approved by the CIPE Resolution 2/2011 of 23.03.2011).

This study has been conducted using E.U. Copernicus Marine Service Information

10    We thank Professor John Huthnance and the Referees for the interest in our study and for the helpful corrections that have greatly improved the work.

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
