# Peer review of "Mesoscale Eddies in the Algerian Basin: do they differ as a function of their formation site?"

_Ocean Science, 2017_

## Referee Comment (RC1) · Anonymous Referee #1 · 9 Mar 2018

The paper makes use of altimeter data (delayed time gridded maps of SLA distributed by CMEMS) from 1993 to 2015 to track eddies in the Algerian Basin. For tracking, the authors implemented an hybrid tracking algorithm spawn from the work of Halo et al. 2014 adapted to the region and with some modifications. The authors then were able to classify sectors within the Algerian basin based on eddy generations/depletions, preferred tracks and so on, evidencing a differences between the eddies associated to the Algerian current and the eddies associated to the North Balearic front. Overall the paper has a clear logical flow and the conclusions (but one statement) reflects the results presented. The title is OK.

The paper however is substantially descriptive, as processes behind the formation/depletion of eddies in the sectors considered, are not tackled. At the same time,

the paper is not methodological, as the methodology implemented is substantially based on previous literature. These considerations somewhat degrades the relevance of the paper for the international community.

Major remarks

Section 2: The tracking algorithm spawn from the work of Halo et al., 2014, while modifications by Pessini et al. are described in section 2.5. The description of the tracking algorithm is very long (also considering that, say 90%, was developed by somebody else) and many details that can be skipped pointing to the existing literature. I also suggest to move regional values adopted and other in-depth details to an appendix to enhance readability.

Section 2.5: In order to prove robustness of the "eddy continuity routine", the authors discuss the successful example of the eddy described by Cotroneo et al. 2016. I wonder if there are cases of failure of this routine and why. Also, I believe the authors should discuss what is the impact of this routine on the results presented later. In the conclusions the authors state that this modification is an improvement, but I cannot judge. In general, there is no attempt by the authors to provide a measure of uncertainty of the methodology.

Results: my main concern is about the threshold between short-life and long-life eddies (90 days). The choice of the threshold does impact the results presented (in particular figs.8-10-11-13 and associated conclusions). The choice of 90 days seems arbitrary and can alter the inference on "longer-life" short life eddies or "shorter-life" long life eddies. To make the analysis robust, I really think the choice of the threshold should be at least inferred from statistical or dynamical arguments.

The reasons behind discrepancies with Escudier et al., 2016 should be discussed.

Minor and editorial remarks:

The dataset used (SLA) should be presented in a separate sub-section of the MM

section, not buried inside the descriptions of the tracking algorithm. The authors specify that the dataset begin in 1993, but not the end (2015?).

Pg 3, l9: Instead of fusco et al 2008, a better references can be Rixen et al GRL 2005 and Schroeder et al SciRep 2016. Also, as detailed in Schroeder et al., 2016, WMDW experiences relatively "short-term" (few years) changes, not long-term (decadal) only. Budillon et al 2009 is grey literature. I would consider dropping it.

Pg4 last para: physical vs. geometrical methods. Many references are listed for physical methods, while none for geometrical methods . . .

Pg 6, l 6: "the number of tunable parameters is thus reduced to three". This statement comes out of the blu

Conclusions, pg 20 l10: "(15)"????

Conclusions, Pg 22 l3-5: I do not agree with the downgrading of the relevance of short-life eddies. Unless the authors support this sentence with some references, I would just erase this statement.

The last paragraph in the conclusion section ("In the past [. . .]") is not justified by any result presented and should be dropped.

Figure 8, 11: If I understood correctly, blue and green bars include also eddies formed and terminated in the same sector (equivalent to yellow bars). My suggestion would be to show in blue only eddies formed in the sector and terminated in a different sector and accordingly for green. In this case, the figure would clearly visualize the dominance of the yellow bar in fig 8 at least.

Font size in figures in general should be made larger.

Fig 10 and 13 are not very high quality figures and results can be easily summarized in one single table instead.

Figure 1: missing many geographical names as well as oceanographic features (e.g.,

Gibraltar, AC, AG, NBF...). All names cited in the text have to be presented in figure 1 for readers unfamiliar with the region.

The way references are managed in this draft may be an academic example on how NOT manage citations and bibliography. I strongly recommend the authors to read OS citation guidelines (https://www.ocean-science.net/Copernicus_Publications_Reference_Types.pdf ) and, why not, to give a try to one of the reference management software available on the market ...

(a) Many places in the text: citations should not include authors' first names (e.g., Isern-Fontanet and E. Garcia-Ladona 2003, Pasquero and A. Provenzale, 2001). Besides, the latter should be Pasquero et al . since the authors are three...

(b) Pinardi et al., 2013 is indeed 2015

(c) Pg 5. L13-14 and Pg 6, l9: websites should not be included in the main text, while listed in the references following OS rules

(d) Penven and Echevin 2005: there 3 missing authors. Accordingly, in the text it should be cited as Penven et al....

(e) Volume number is generally not mandatory, but page numbers are.

(f) Puillat et al 2002: is the title incorrect?

(g) Pasquero et al. 2001. As said, missing one author (A. B. should be A. Babiano...)

I may have missed other errors...

———————————————————

---

## Referee Comment (RC2) · Anonymous Referee #2 · 14 Mar 2018

**Review on the manuscript**

**"Mesoscale Eddies in the Algerian Basin: do they differs as a function of their formation site ?"**

**by F. Pessini, A. Olita, Y. Cotroneo and A. Perilli**

**submitted to Ocean Science**

This work presents a statistical analysis of the dynamics of meso scale anticyclones in the western mediterranean sea, and especially in the Algerian Basin, performed with an automatic eddy detection and tracking algorithm applied to 22 years sea level anomaly of the AVISO/DUACS data set. This analysis emphasis on the dynamical properties of two distinct types of anticylones, the AEs and the FAEs, formed in distinct area. However, due to some a priori choices which exclude the cyclones from the analysis or limit the area of investigation, some important dynamical features are missed in the present analysis. I listed below a numbers of important issues that should be satisfactorily addressed in order to consider a (major) revised paper for publication in Ocean Science.

**Major comments:**

**1. Sensitivity of the eddy contour to the Okubo-Weiss threshold W.**

Several studies (Sadarjoen and Post 2000; Chaigneau et al. 2008; LeVu et al. 2017) mentionned a high sensitivity of the size and the shape of the detected eddies to the threshold value W and a general tendency for false positive eddy detection. Isern-Fontanet *et al.* (2003, 2005) suggests to use the threshold  $W = -0.2\sigma$  to identify the vortex cores, where  $\sigma$  is the standard deviation of the W distribution among the domain. Another study (Chaigneau et al. 2008) suggests that the best compromise is a value of W in the range  $-0.3\sigma \leq W \leq -0.2\sigma$  while Chelton et al. (2007), propose to use a fixed value  $W = 2.10^{-12}s^{-2}$  for the eddy detection. In the present study, the authors fixed the threshold value W = 0. However, it is never explained why? The authors, shoud at least investigate how the typical eddy size and the EKE is affected if this threshold value is changed? The method limitations are brought up in the manuscript but never quantified, for instance what is the sensitivity of their eddy detection to the Okubo-Weiss threshold W?

**2**. The statistical analysis of cyclonic eddies is missing.**

The authors restrict their statistical analysis to anticyclonic eddies, assuming that most of the cyclonic eddies are short-lived. First, they should confirm that, according to their eddy detection and tracking algorithm, this is indeed the case ! Besides, a significant part of their analysis focus on short-lived anticyclones (shorter than 90 days). I would be very surprize if they do not find a large fraction of cyclonic eddies for this range of lifetime. Mkhinini *et al.* (2014) have shown, in the eastern mediterranean basin, that it is only when the lifetime exceeds 6 months that the anticyclones become dominant. Hence, even if their lifetime are shorter, cyclonic eddies could be more numerous than anticyclones and contribute significantly to the eddy kinetic energy. The recent work of Escudier et al. (2016) investigate AE's of both sign and found that their propagation speed differs. Therefore, a significant fraction of mesoscale eddies (the cyclonic ones) are missing in this study and should be investigated to assess correctly the EKE distribution in the Algerian basin. The terms "eddies" or "eddy" used throughout the manuscript is really misleading because it always correspond to anticyclones (i.e. A.E.).

**3.** The western part of the Algerian basin $(\langle 2^{\circ}E \rangle)$ is not studied.**

The authors restrict their analysis into an area that does not extend below 2°E. However, the previous study of Escudier et al. (2016) have shown that Algerian eddies are , on average, advected along two large cyclonic loops. The first one is located between 0°-4°E and the second one between 5°E-8°E. Besides, according to the Figure 11 of Escudier et al. (2016), three main formation areas are located along the Algerian coast: 1°W-0°E, 1°E-3°E and 5°E-8°E. Hence, the limit of 2°E used in the present study exclude two main formation areas of AE and cut the first cyclonic loop which characterize the trajectories of long-lived eddies. The statistical analysis of the eddy formation and termination in the box D will be strongly impacted by the westearn part of the basin (1°W-2°E) which is unfortunately not studied in the paper. Therefore, a larger domain should be investigated to describe accurately the spatial and temporal distribution of long-lived eddies in the Algerian Basin.

**4.** *Interactions and transport between FAEs and AEs.**

One conclusion of this paper is to emphasis that the Algerain basin can be separated in two parts, the Algerian coast and the Balearic Front, with (almost) no connections between these two area. However, there is a striking correlation in the time series (Figure 6) of the anticyclonic EKE between the southern Algerian basin and the northern basin. The three peaks that occur in the northern part seems to be correlated (6 months shift) to the three peaks of the southern part. Besides, according to the eddy trajectories shown in the figures 12 and 14 some long-lived anticyclones formed along the Algerian coast crosses the 39°N latitude and may therefore interacts with the North Balearic Front or the FAEs. These two types of eddies could merge together in the central part of the basin. Therefore, the statement of "no-connection" or "no-interarctions" between AEs and FAEs seems doubtfull. I encourage the authors to investigate more carfully the possible interactions between the AEs and the FAEs rather than emphasis on a virtual "separation" between the north and the south.

**Other comments:**

**5.** page 2 line 13: The following references are mainly related to the western Mediterranean Sea (WMED), therefore the authors should be more explicit here and mention "Mesoscale eddies in the **western** Mediterranean sea have been widely investigated in the past...". Otherwise, many other papers related to the eastern meditteranenan eddies should also be mentionned.

**6.** page 2 line 15. Some recents papers related to in-situ measurements of meso scale eddies in the Western mediterranean sea, especially from glider survey, should be mentionned here:

- Amores, et al. (2013), J. Geophys. Res. Oceans, doi:10.1002/jgrc.20150.
- Cotroneo, et al. Journal of marine systems. (2016).
- G.Aulicino et al. Journal of marine systems (2018). (https://doi.org/10.1016/j.jmarsys.2017.11.006).

The following reference, related to the intercomparisons between satellite altimetry and numerical model, is also missing:

- Escudier, et al. (2016), J. Geophys. Res. Oceans, 121, doi:10.1002/2015JC011371.

7. page 2 line 16: "Most data on the motion of the eddies are provided by infrared and colour satellite imagery" I am a bit suprized by such statement, because the visible images do not provides any quantitative informations on the intensity (velocity or vorticity) of the detected eddies. Besides, as far as I know, there is no automatic methods or algorithm able to track the eddies on visible images. The number of eddy trajectories deduced from infrared or colour satellite images remain limited and subject to qualitative interpretation. It is therefore very difficult to get any statistical analysis on the dynamics or even the drifting speed of meso scale eddies only from visible images. **8**. page 5 line1-2: The automatic eddy detection algorithms based on geometrical methods or hybrid ones are not fully explained and some appropriate references are missing here. Sadarjoen and Post (2000) and Nencioli et al. (2010) used only the geometrical properties of closed streamlines to identify coherent vortices regardless of their intensity. As far as hybrid methods are concerned, some studies (Viikmäe and Torsvik 2013; Halo et al. 2014; Yi et al. 2014) used the OW parameter to detect the possible eddy centers, while Mkhinini et al. (2014) and LeVu et al. (2017) used the local normalized angular momentum (LNAM).

**9.** page 12 line 21-27: The figure 4 of this paper should be compared to the figure 9(a) of Escudier et al. (doi:10.1002/2015JC011371.) which shows similar patterns of the eddy density in the Algerian Basin.

**10**. The authors often give very precise numbers "A total of 125,256 anticyclonic eddies and 127,761 cyclonic eddies were detected" or values with two digits "The mean radius of anticyclonic (cyclonic) eddies was 97.78 (96.53) km". I am not sure that the eddy detection algorithm is so precise and accurate !! a rough order of magnitude will be sufficient, such as "125,000 anticyclonic eddies were detected" and "a mean radius around 98 km"...

**11**. Page 20, the trajectory of the AE depicted in Figure 14(a) should be compared in more details with the one deduced from the analysis of SST images in Puillat et al. (2002). In the latter, this long-lived anticyclone was detected up to December 1998 and not November 1997. Besides, the termination point is located at 1.5°E and not around 8°W. The differences between these distinct trajectories of the same eddy should be discussed.

**References:**

- Amores, A., S. Monserrat, and M. Marcos (2013), Vertical structure and temporal evolution of an anticyclonic eddy in the Balearic Sea (western Mediterranean), J. Geophys. Res. Oceans, 118, 2097–2106, doi:10.1002/jgrc.20150.

- Le Vu, B., Stegner, A., & Arsouze, T. (2017). Angular Momentum Eddy Detection and tracking Algorithm (AMEDA) and its application to coastal eddy formation. DOI: 10.1175/JTECH-D-17-0010.1

- Mkhinini, N., A. L. S. Coimbra, A. Stegner, T. Arsouze, I. Taupier- Letage, and K. Béranger, (2014): Long-lived mesoscale eddies in the eastern Mediterranean Sea: Analysis of 20 years of AVISO geostrophic velocities. J. Geophys. Res. Oceans, 119, 8603–8626, https://doi.org/10.1002/2014JC010176.

- Nencioli, F., C. Dong, T. Dickey, L. Washburn, and J. C. McWilliams, 2010: A vector geometry–based eddy detection algorithm and its application to a high-resolution numerical model product and high-frequency radar surface velocities in the Southern California Bight. J. Atmos. Oceanic Technol., 27, 564–579, https://doi.org/10.1175/2009JTECHO725.1.

- Sadarjoen, I. A., and F. H. Post, 2000: Detection, quantification, and tracking of vortices using streamline geometry. Comput. Graphics, 24, 333–341, https://doi.org/10.1016/S0097- 8493(00)00029-7.

- Viikmäe, B., and T. Torsvik, 2013: Quantification and character- ization of mesoscale eddies with different automatic identifi- cation algorithms. J. Coastal Res., 65, 2077–2082, https://doi. org/10.2112/SI65-351.1.

---

## Author Comment (AC1) · 9 Apr 2018

Reviewer #1

The paper makes use of altimeter data (delayed time gridded maps of SLA distributed by CMEMS) from 1993 to 2015 to track eddies in the Algerian Basin. For tracking, the authors implemented an hybrid tracking algorithm spawn from the work of Halo et al. 2014 adapted to the region and with some modifications. The authors then were able to classify sectors within the Algerian basin based on eddy generations/ depletions, preferred tracks and so on, evidencing a differences between the eddies associated to the Algerian current and the eddies associated to the North Balearic front. Overall the paper has a clear logical flow and the conclusions (but one statement) reflects the results presented. The title is OK.
The paper however is substantially descriptive, as processes behind the formation/ depletion of eddies in the sectors considered, are not tackled. At the same time, the paper is not methodological, as the methodology implemented is substantially based on previous literature. These considerations somewhat degrades the relevance of the paper for the international community.

We thank the Referee for considering our paper and for raising a number of important points. All the corrections resulting from his/her comments will be included in the final version of the manuscript. As well, as requested, we will describe the processes linked to the formation and depletion of eddies. Additionally, the differences between the applied tracking method and the methods published in previous literature have been better described.

**Major remarks:**

Section 2: The tracking algorithm spawn from the work of Halo et al., 2014, while modifications by Pessini et al. are described in section 2.5. The description of the tracking algorithm is very long (also considering that, say 90%, was developed by somebody else) and many details that can be skipped pointing to the existing literature. I also suggest to move regional values adopted and other in-depth details to an appendix to enhance readability.

We will reduce the description of the tracking algorithm, moving some details to an appendix as suggested.

Section 2.5: in order to prove robustness of the "eddy continuity routine", the authors discuss the successful example of the eddy described by Cotroneo et al. 2016. I wonder if there are cases of failure of this routine and why. Also, I believe the authors should discuss what is the impact of this routine on the results presented later. In the conclusions the authors state that this modification is an improvement, but I cannot judge. In general, there is no attempt by the authors to provide a measure of uncertainty of the methodology.

The tracking algorithm is unable to recognise events such as merging and bifurcation of eddies, and the continuity routine does not fix this problem but at the same time it does not add errors to the results.

An evidence of failure would be represented by mistakenly joining of two structures, which are not easily detectable by using SLA data. In fact, we did not find any such case.

In our opinion, a good example of the mode of operation of the continuity routine can be shown by the "very long-life eddy" observed by Puillat et al., 2001. The tracking algorithm, even after application of the continuity routine, erroneously detects the termination of the structure near the coast, where the closed contour of Okubo-Weiss bifurcates. Nevertheless, in this case the continuity routine proves to be useful as it joins two separated features (410 and 295 days long respectively) into a single structure 706 days long.

The cases of bifurcation and merging of the structures are complex topics we have not addressed in this work, but which will be the object of detailed study in future. Nevertheless, the tracking algorithm is not the most suitable tool to investigate the interaction between these structures.

In conclusion, we can affirm that the continuity routine we have implemented improves the results, even if it does not resolve the problems of merging and bifurcation of eddies.

The application of the continuity routine over 24 years leads to the decrease of the total number of eddies detected (from 8208 to 6543) with the consequent increase of the mean lifespan from around 66 to 88 days.

Results: my main concern is about the threshold between short-life and long-life eddies (90 days). The choice of the threshold does impact the results presented (in particular figs.8-10-11-13 and associated conclusions). The choice of 90 days seems arbitrary and can alter the inference on "longer-life" short life eddies or "shorter-life" long life eddies. To make the analysis robust, I really think the choice of the threshold should be at least inferred from statistical or dynamical arguments.

The threshold value of 90 days has been chosen as a function of the distribution of all the lifetimes of anticyclonic eddies. In fact, north of 39° N, the lifetimes within the 97[th] percentile are shorter than 90 days. Furthermore, we observed that 97% of the cyclonic features (included on the suggestion of Reviewer #2) in the northern part of the basin have lifespans shorter than 90 days.

In general, considering the whole basin, 95% of both kinds of structures have lifespans less than 90 days.

For these reasons we would maintain the threshold value of 90 days to discriminate between short- and long-life eddies.

The reasons behind discrepancies with Escudier et al., 2016 should be discussed.

The general intention of our manuscript is to highlight the differences between the structures formed along the Algerian Current and the eddies formed in proximity to the North Balearic Front, while the paper of Escudier et al., 2016 focuses only on the

southern Algerian Eddies. Nevertheless, a further description of differences and similarities with the paper of Escudier et al., 2016 will be inserted in the new version of the manuscript.
In particular, as the new version will include the analysis of cyclonic and anticyclonic structures over a larger spatial domain, it will ease the comparison.

**Minor and editorial remarks:**

The dataset used (SLA) should be presented in a separate sub-section of the MM section, not buried inside the descriptions of the tracking algorithm. The authors specify that the dataset begin in 1993, but not the end (2015?).

We changed the structure of the draft as suggested.
SLA data are now available until the end of 2016 (2014 in the previous version). In the new version, we applied the tracking method and elaborated the statistics to the end of 2016.

Pg 3, l9: Instead of fusco et al 2008, a better references can be Rixen et al GRL 2005 and Schroeder et al SciRep 2016. Also, as detailed in Schroeder et al., 2016, WMDW experiences relatively "short-term" (few years) changes, not long-term (decadal) only. Budillon et al 2009 is grey literature. I would consider dropping it.

We changed and integrated the references as suggested.

Pg4 last para: physical vs. geometrical methods. Many references are listed for physical methods, while none for geometrical methods…

We added references for geometrical methods, and in particular Sadarjoen and Post, 2000 and Nencioli et al., 2010 were usefully inserted.

Pg 6, l 6: "the number of tunable parameters is thus reduced to three". This statement comes out of the blu.

The statement was not clear, so we will better explain which parameters in the detection algorithm can be changed according to the researcher's interests.
In particular this sentence has been changed as follows:
 "The tunable parameters in the detection algorithm are three: the interval between the contours of SLA, the maximum radius of a closed contour of SLA detected and the threshold of the Okubo-Weiss parameter.

Conclusions, pg 20 l10: "(15)"????

It indicates the number of the equation, but "Eq." was missing. Corrected.

Conclusions, Pg 22 l3-5: I do not agree with the downgrading of the relevance of shortlife eddies. Unless the authors support this sentence with some references, I would just erase this statement.

The sentence has been removed.

The last paragraph in the conclusion section ("In the past […]") is not justified by any result presented and should be dropped.

The paragraph has been removed

Figure 8, 11: If I understood correctly, blue and green bars include also eddies formed and terminated in the same sector (equivalent to yellow bars). My suggestion would be to show in blue only eddies formed in the sector and terminated in a different sector and accordingly for green. In this case, the figure would clearly visualize the dominance of the yellow bar in fig 8 at least.

We agree with the observation of the Reviewer. In the hope of making the graph clearer we have included the bars of "formation and vanishing" over the bars of the "formation".

Font size in figures in general should be made larger.

We think that it depends on the Copernicus template, as we used the style suggested in the Latex file, but we will check and eventually we will ask to enlarge the font size in all the figures.

Fig 10 and 13 are not very high quality figures and results can be easily summarized in one single table instead.

We will substitute the figures with a single table as suggested.

Figure 1: missing many geographical names as well as oceanographic features (e.g., Gibraltar, AC, AG, NBF…). All names cited in the text have to be presented in figure 1 for readers unfamiliar with the region.

We will add more geographical names in order to better describe the study area.

The way references are managed in this draft may be an academic example on how NOT manage citations and bibliography. I strongly recommend the authors to read OS citation guidelines (https://www.oceanscience. net/Copernicus_Publications_Reference_Types.pdf ) and, why not, to give a try to one of the reference management software available on the market…
(a) Many places in the text: citations should not include authors' first names (e.g., Isern- Fontanet and E. Garcia-Ladona 2003, Pasquero and A. Provenzale, 2001). Besides, the latter should be Pasquero et al . since the authors are three…
(b) Pinardi et al., 2013 is indeed 2015
(c) Pg 5. L13-14 and Pg 6, l9: websites should not be included in the main text, while listed in the references following OS rules
(d) Penven and Echevin 2005: there 3 missing authors. Accordingly, in the text it should be cited as Penven et al….
(e) Volume number is generally not mandatory, but page numbers are.
(f) Puillat et al 2002: is the title incorrect?
(g) Pasquero et al. 2001. As said, missing one author (A. B. should be A. Babiano…)
I may have missed other errors…

The bibliography will be modified according to the correct format.

---

## Author Comment (AC2) · 9 Apr 2018

Reviewer #2

This work presents a statistical analysis of the dynamics of meso scale anticyclones in the western mediterranean sea, and especially in the Algerian Basin, performed with an automatic eddy detection and tracking algorithm applied to 22 years sea level anomaly of the AVISO/DUACS data set. This analysis emphasis on the dynamical properties of two distinct types of anticyclones, the AEs and the FAEs, formed in distinct area. However, due to some a priori choices which exclude the cyclones from the analysis or limit the area of investigation, some important dynamical features are missed in the
present analysis. I listed below a numbers of important issues that should be satisfactorily addressed in order to consider a (major) revised paper for publication in Ocean Science.

We thank the Referee for the interest in our work, for the accurate review and for the helpful corrections.
We have checked all the major and minor comments and have made necessary changes accordingly to the indications provided.
Below, we address each comment point by point.

Major comments:

*1) Sensitivity of the eddy contour to the Okubo-Weiss threshold $W_0$.*
Several studies (Sadarjoen and Post 2000; Chaigneau et al. 2008; LeVu et al. 2017) mentioned a high sensitivity of the size and the shape of the detected eddies to the threshold value W and a general tendency for false positive eddy detection. Isern-Fontanet et al. (2003, 2005) suggests to use the threshold $W = -0.2\ \sigma$ to identify the vortex cores, where $\sigma$ is the standard deviation of the W distribution among the domain. Another study (Chaigneau et al. 2008) suggests that the best compromise is a value of W in the range $-0.3\ \sigma \le W \le -0.2\ \sigma$ while Chelton et al. (2007), propose to use a fixed value
$W = 2.10^{-12} s^{-2}$ for the eddy detection. In the present study, the authors fixed the threshold value $W = 0$. However, it is never explained why? The authors, should at least investigate how the typical eddy size and the EKE is affected if this threshold value is changed? The method limitations are brought up in the manuscript but never quantified, for instance what is the sensitivity of their eddy detection to the Okubo-Weiss threshold W?

We thank the referee for this comment, as it offers the opportunity to describe the sensitivity test over the entire domain (in space and time) and to better explain our choice of $W_0=0$.
Isern-Fontanet et al., 2004 define the eddy core as the area where the Okubo-Weiss parameter is less than $W_0=-0.2\ \sigma_0$, where $\sigma_0$ is the standard deviation in the whole domain, i.e. the Mediterranean Sea. The values of $\sigma_0$ are of the order of $10^{-11}$ according to their paper. In the new version of our manuscript, we describe the results of the sensitivity test led over 24 years (1993-2016) within the new (enlarged) domain with $W_0 = 0$ and $W_0 = -2.1*10^{-12}$ (according to Chelton 2007 and very close to the values suggested by Isern-Fontanet). The results show that there is a decrease of 0.4%

in the number of anticyclones and an increase of 0.2% in the number of the cyclones detected in the area of study using the latter value.

The variation of the mean radius is around 3% (smaller with $W_0 = -2.1*10^{-12}$) for both cyclones and anticyclones.

The energy variation, using $W_0=-2.1*10^{-12}$, is 0.4% smaller for anticyclonic structures and 0.2% smaller for the cyclonic ones.

Despite these minor discrepancies in the mean values, some differences arise when looking at the specific properties of some well-known structures.

In particular, we observed that in the case of $W_0=0$ the detection method (and consequently the tracking method) identifies a structure characterised by a lifetime of 610 days formed in the southern part of the basin. The same structure is unfortunately not well detected setting $W_0 = -2.*10^{-12}$. In fact, with the latter value, the same structure has a lifespan of 178 days due to the misleading detection of three closed Okubo-Weiss contours (trifurcation). The comparison with the daily SLA maps suggests that the choice of $W_0 = 0$ in this case is better suited.

Furthermore, setting $W_0=-2.1*10^{-12}$ the algorithms tracked a structure 466 days long in the northern part of the basin, which appears shorter (296 days) with $W_0 = 0$. In this case, the analysis of daily SLA maps suggests that the use of $W_0 =0$ is preferable.

It is important to note that the peaks of energy, which find correspondence in the "very long-life" tracks (> 450 days), would not have a corresponding "very long-life" track in 2013 setting $W_0=-2.1*10^{-12}$.

In general, our idea is that the smaller value of $W_0$ highlights the daily presence of weak and small structures that could create "noise", and occasionally influence the detection of longer and stronger features.

For these reasons we opted for $W_0=0$, and this choice is now better described in the manuscript.

*2) The statistical analysis of cyclonic eddies is missing.*
The authors restrict their statistical analysis to anticyclonic eddies, assuming that most of the cyclonic eddies are short-lived. First, they should confirm that, according to their eddy detection and tracking algorithm, this is indeed the case ! Besides, a significant part of their analysis focus on short-lived anticyclones (shorter than 90 days). I would be very surprize if they do not find a large fraction of cyclonic eddies for this range of lifetime. Mkhinini et al. (2014) have shown, in the eastern mediterranean basin, that it is only when the lifetime exceeds 6 months that the anticyclones become dominant. Hence, even if their lifetime are shorter, cyclonic eddies could be more numerous than anticyclones and contribute significantly to the eddy kinetic energy. The recent work of Escudier et al. (2016) investigate AE's of both sign and found that their propagation speed differs. Therefore, a significant fraction of mesoscale eddies (the cyclonic ones) are missing in this study and should be investigated to assess correctly the EKE distribution in the Algerian basin. The terms "eddies" or "eddy" used throughout the manuscript is really misleading because it always corresponds to anticyclones (i.e. A.E.).

We agree with the Referee on the importance of cyclonic eddies, so their analysis will be included in the revised manuscript.

To summarise some of the results that will be included in the revised manuscript: cyclonic structures longer than 7 days number 3,838 (a greater quantity than the

anticyclones, as suggested by the Reviewer), and the mean lifetime is around 21 days (shorter than the mean lifetime of the anticyclones). 95% of the structures have a lifetime shorter than 65 days (while 95% of the anticyclones have a lifetime shorter than 90 days). The maximum lifetime of the cyclones is 314 days and this single case will be studied and described in the manuscript. Despite their bigger number, the mean kinetic energy of cyclones is half the mean kinetic energy of the anticyclones. Over 24 years we estimate the total amount kinetic energy of cyclones to be some 30% lower than that of anticyclones due to the larger number of the former.

*3) The western part of the Algerian basin (<2°E) is not studied.*
The authors restrict their analysis into an area that does not extend below 2°E. However, the previous study of Escudier et al. (2016) have shown that Algerian eddies are, on average, advected along two large cyclonic loops. The first one is located between 0°-4°E and the second one between 5°E-8°E. Besides, according to the Figure 11 of Escudier et al. (2016), three main formation areas are located along the Algerian coast: 1°W-0°E, 1°E-3°E and 5°E-8°E. Hence, the limit of 2°E used in the present study exclude two main formation areas of AE and cut the first cyclonic loop which characterize the trajectories of long-lived eddies. The statistical analysis of the eddy formation and termination in the box D will be strongly impacted by the westearn part of the basin (1°W-2°E) which is unfortunately not studied in the paper. Therefore, a larger domain should be investigated to describe accurately the spatial and temporal distribution of long-lived eddies in the Algerian Basin.

As suggested, the area of study has now been enlarged to include all the spawning area of the Algerian Eddies. The new domain extends from 2°W to 11° E and from 37° to 42° N. We are calculating the statistics over the new domain and a more detailed comparison with the results of Escudier et al, 2016 will be discussed (as suggested by Referee #1 too).

*4) Interaction and transport between FAEs and AEs*
One conclusion of this paper is to emphasis that the Algerain basin can be separated in two parts, the Algerian coast and the Balearic Front, with (almost) no connections between these two area. However, there is a striking correlation in the time series (Figure 6) of the anticyclonic EKE between the southern Algerian basin and the northern basin. The three peaks that occur in the northern part seems to be correlated (6 months shift) to the three peaks of the southern part. Besides, according to the eddy trajectories shown in the figures 12 and 14 some long-lived anticyclones formed along the Algerian coast crosses the 39°N latitude and may therefore interacts with the North Balearic Front or the FAEs. These two types of eddies could merge together in the central part of the basin. Therefore, the statement of "noconnection" or "no-interarctions" between AEs and FAEs seems doubtfull. I encourage the authors to investigate more carfully the possible interactions between the AEs and the FAEs rather than emphasis on a virtual "separation" between the north and the south.

This suggestion is very helpful and we thank the Referee for having raised the point.

The main aim of the work is to highlight the difference between the mesoscale structures forming along the Algerian coast and the features forming in proximity to the North Balearic Front. In spite of the similar processes of formation, we note differences in their kinetic energy and pathways.

As suggested by the Referee, we investigated the correlations between the two time series in Figure 6 in order to find the possible connections between the northern and the southern areas.

The peaks of energy in the northern part in 2006 and between 2008 and 2010 (shifted by about 6 months with respect to the peaks of the southern part) can be correlated with the crossing of the 39$^{th}$ parallel N by the "very-long life" eddies formed in the southern part of the basin.

In fact, the mean kinetic energy of these southern features is about three times higher than the mean EKE of the northern ones. According to this estimation, even the transition of a single long-life eddy from the southern to the northern part of the basin can have a remarkable effect on the monthly mean EKE of the area.

The first EKE peak in the northern part (1998-1999) seems not to be related to structures in the southern part, but it will be more carefully studied.

The interactions between eddies (merging) or with the coast have to be further investigated through additional tools, as the tracking routine is unable for the time being to recognise these kinds of events.

Finally, thanks to the suggestions received from the Referees, we are now designing a connectivity analysis among several sub-areas in the Western Mediterranean basin. By examining the eddy pathways through the different sub-areas, we intend to compute the transit and residence time, pseudo-Eulerian statistics and connection probabilities as successfully performed in several studies on Lagrangian drifters.

Minor comments

5. page 2 line 13: The following references are mainly related to the western Mediterranean Sea (WMED), therefore the authors should be more explicit here and mention "Mesoscale eddies in the western Mediterranean sea have been widely investigated in the past...". Otherwise, many other papers related to the eastern Mediterranean eddies should also be mentioned.

Sentence corrected.

6. page 2 line 15. Some recents papers related to in-situ measurements of meso scale eddies in the Western mediterranean sea, especially from glider survey, should be mentionned here:
-Amores, et al. (2013), J. Geophys. Res. Oceans, doi:10.1002/jgrc.20150.
-Cotroneo, et al. Journal of marine systems. (2016).
-G.Aulicino et al. Journal of marine systems (2018). (https://doi.org/10.1016/j.jmarsys.2017.11.006).
The following reference, related to the inter-comparisons between satellite altimetry and numerical model, is also missing:
-Escudier, et al. (2016), J. Geophys. Res. Oceans, 121, doi:10.1002/ 2015JC011371.

References added or better located than in the previous version.

7. page 2 line 16: "Most data on the motion of the eddies are provided by infrared and colour satellite imagery" I am a bit surprized by such statement, because the visible images do not provides any quantitative informations on the intensity (velocity or vorticity) of the detected eddies. Besides, as far as I know, there is no automatic methods or algorithm able to track the eddies on visible images. The number of eddy trajectories deduced from infrared or colour satellite images remain limited and subject to qualitative interpretation. It is therefore very difficult to get any statistical analysis on the dynamics or even the drifting speed of meso scale eddies only from visible images.

We corrected "Most data on the motion…" with "Most of qualitative information on the motion…".

8. page 5 line1-2: The automatic eddy detection algorithms based on geometrical methods or hybrid ones are not fully explained and some appropriate references are missing here. Sadarjoen and Post (2000) and Nencioli et al. (2010) used only the geometrical properties of closed streamlines to identify coherent vortices regardless of their intensity. As far as hybrid methods are concerned, some studies (Viikmäe and Torsvik 2013; Halo et al. 2014; Yi et al. 2014) used the OW parameter to detect the possible eddy centers, while Mkhinini et al. (2014) and LeVu et al. (2017) used the local normalized angular momentum (LNAM).

We thank the referee for his/her indications. New references for geometrical methods have been added.

9. page 12 line 21-27: The figure 4 of this paper should be compared to the figure 9(a) of Escudier et al. (doi:10.1002/ 2015JC011371.) which shows similar patterns of the eddy density in the Algerian Basin.

A larger and more detailed comparison with the paper of Escudier et al., 2016 is now performed throughout the manuscript, also on the basis of new results deriving from the modification of the spatial domain of our study.
In particular, we will add some considerations to clarify the comparison with Escudier's figure.

10. The authors often give very precise numbers "A total of 125,256 anticyclonic eddies and 127,761 cyclonic eddies were detected" or values with two digits "The mean radius of anticyclonic (cyclonic) eddies was 97.78 (96.53) km". I am not sure that the eddy detection algorithm is so precise and accurate!! a rough order of magnitude will be sufficient, such as " 125,000 anticyclonic eddies were detected "

and "a mean radius around 98 km"...

All numbers have now been standardized to a more representative number of digits. Corrected.

11. Page 20, the trajectory of the AE depicted in Figure 14(a) should be compared in more details with the one deduced from the analysis of SST images in Puillat et al. (2002). In the latter, this long-lived anticyclone was detected up to December 1998 and not November 1997. Besides, the termination point is located at 1.5°E and not around 8°W. The differences between these distinct trajectories of the same eddy should be discussed.

We will discuss in more depth the differences between our method and that applied by Puillat et al. (2002) in the tracking of the same structure.
In our opinion, the differences depend mostly on the different kinds of data being analysed. In fact, the tracks suggested by Puillat et al., 2002 has been computed over SST images and weekly SLA data. In particular, the authors declared the date of formation of the eddy uncertain due to the unavailability of some data.
Furthermore, differences in the termination place and date may be linked to the difficulty of the detecting algorithm to identify structures near the coast.
These differences are now explained and detailed in the manuscript together with our hypothesis.

References:
- Amores, A., S. Monserrat, and M. Marcos (2013), Vertical structure and temporal evolution of an anticyclonic eddy in the Balearic Sea (western Mediterranean), J. Geophys. Res. Oceans, 118, 2097–2106, doi:10.1002/jgrc.20150.
- Le Vu, B., Stegner, A., & Arsouze, T. (2017). Angular Momentum Eddy Detection and tracking Algorithm (AMEDA) and its application to coastal eddy formation. DOI: 10.1175/JTECH-D-17-0010.1
- Mkhinini, N., A. L. S. Coimbra, A. Stegner, T. Arsouze, I. Taupier- Letage, and K. Béranger, (2014): Long-lived mesoscale eddies in the eastern Mediterranean Sea: Analysis of 20 years of AVISO geostrophic velocities. J. Geophys. Res. Oceans, 119, 8603–8626, https://doi.org/10.1002/2014JC010176.
- Nencioli, F., C. Dong, T. Dickey, L.Washburn, and J. C. McWilliams, 2010: A vector geometry–based eddy detection algorithm and its application to a high-resolution numerical model product and highfrequency radar surface velocities in the Southern California Bight. J. Atmos. Oceanic Technol., 27, 564–579, https://doi.org/10.1175/2009JTECHO725.1.
- Sadarjoen, I. A., and F. H. Post, 2000: Detection, quantification, and tracking of vortices using streamline geometry. Comput. Graphics, 24, 333–341, https://doi.org/10.1016/S0097- 8493(00)00029-7.
- Viikmäe, B., and T. Torsvik, 2013: Quantification and character- ization of mesoscale eddies with different automatic identification algorithms. J. Coastal Res., 65, 2077–2082, https://doi. org/10.2112/SI65-351.1.

We thank the Referee for suggesting these helpful references, which have been added to the citation list.